



# Street-scale air quality modelling for Beijing during a winter 2016 measurement campaign

Michael Biggart[1], Jenny Stocker[2], Ruth M. Doherty[1], Oliver Wild[3], Michael Hollaway[3,12], David Carruthers[2], Jie Li[4], Qiang Zhang[5], Ruili Wu[5], Simone Kotthaus[6,7], Sue Grimmond[6], Freya A. Squires[8], James Lee[8,9], and Zongbo Shi[10,11]

[1]School of Geosciences, The University of Edinburgh, Edinburgh, UK

[2]Cambridge Environmental Research Consultants, Cambridge, UK

[3]Lancaster Environment Centre, Lancaster University, Lancaster, UK

[4]State Key Laboratory of Atmospheric Boundary Layer Physics and Atmospheric Chemistry, Institute of Atmospheric Physics, Chinese Academy of Sciences, Beijing, China

[5]Ministry of Education Key Laboratory for Earth System Modelling, Department of Earth System Science, Tsinghua University, Beijing, China

[6]Department of Meteorology, University of Reading, Reading, UK

[7]Institut Pierre Simon Laplace, École Polytechnique, Palaiseau, France

[8]Wolfson Atmospheric Chemistry Laboratories, Department of Chemistry, University of York, York, UK

[9]National Centre for Atmospheric Science, University of York, York, UK

[10]School of Geography Earth and Environmental Sciences, University of Birmingham, Birmingham, UK

[11]Institute of Surface-Earth System Science, Tianjin University, Tianjin, China

[12]Now at: Centre for Ecology & Hydrology, Lancaster Environment Centre, Bailrigg, Lancaster, UK

*Correspondence to:* Michael Biggart (Michael.Biggart@ed.ac.uk)

**Abstract.** We examine the street-scale variation of $NO_x$, $NO_2$, $O_3$ and $PM_{2.5}$ concentrations in Beijing during the Atmospheric Pollution and Human Health in a Chinese Megacity (APHH-China) winter measurement campaign in November - December 2016. Simulations are performed using the urban air pollution dispersion and chemistry model ADMS-Urban, and an explicit network of road source emissions. Two versions of the gridded Multi-resolution Emission Inventory for China (MEIC v1.3) are used: the standard MEIC v1.3 emissions and an optimised version, both at 3 km resolution. We construct a new traffic emissions inventory by apportioning the transport sector onto a detailed spatial road map. Agreement between mean simulated and measured pollutant concentrations from Beijing's air quality monitoring network and the Institute of Atmospheric Physics (IAP) field site is improved when using the optimised emissions inventory. The inclusion of fast $NO_x$-$O_3$ chemistry and explicit traffic emissions enables the sharp concentration gradients adjacent to major roads to be resolved with the model. However, $NO_2$ concentrations are overestimated close to roads, likely due to the assumption of uniform traffic activity across the study domain. Differences between measured and simulated diurnal $NO_2$ cycles suggest that an additional evening $NO_x$ emission source, likely related to heavy duty diesel trucks, is not fully accounted for in the emissions inventory. Overestimates in simulated early evening $NO_2$ are reduced by delaying the formation of stable boundary layer conditions in the model to replicate Beijing's urban heat island. The simulated campaign period mean $PM_{2.5}$ concentration range across the monitoring network (~15 μgm$^{-3}$) is much lower than the measured range (~40 μgm$^{-3}$). This is likely a consequence of insufficient $PM_{2.5}$ emissions and spatial variability, neglect of explicit point sources, and assumption of a homogeneous background $PM_{2.5}$ level. Sensitivity studies highlight that the use of explicit road source emissions, modified diurnal emission profiles, and inclusion of urban heat island effects permit closer agreement between simulated and measured $NO_2$ concentrations. This work lays the foundations for future studies of human exposure to ambient air pollution across complex urban areas, with the APHH-China campaign measurements providing a valuable means of evaluating the impact of key processes on street-scale air quality.


## 1 Introduction

In recent decades, China's rapid economic growth, industrialisation and urbanisation has led to severely deteriorating air quality. Associations between high concentrations of air pollutant species, such as fine particulate matter ($PM_{2.5}$), nitrogen oxides ($NO_x = NO + NO_2$) and ozone ($O_3$), and adverse health effects are well-established in China (Han et al. 2018). Most notably, the inhalation of ambient $PM_{2.5}$ is linked to respiratory illnesses, cardiovascular disease, lung cancer and adverse birth outcomes (Han et al. 2018; Liang et al. 2019). The Global Burden of Disease Study 2016 identified ambient $PM_{2.5}$ exposure as the fourth leading cause of premature death in China (GBD 2016 Risk Factors Collaborators, 2017).

To accurately assess the extent of human exposure to pollution in densely populated and complex urban areas and to reduce this health risk, comprehensive information is needed on the spatiotemporal variation of ambient pollutant concentrations, the dominant emission source sectors, chemical processes and the role of meteorological conditions in pollution accumulation and dispersion. High quality air pollutant concentration measurements can provide some of the required information. For instance in Beijing, a 35 station automated air quality monitoring network has measured continuous hourly concentrations of $PM_{2.5}$, $PM_{10}$, $SO_2$, $NO_2$, $O_3$ and CO since 2013. However, these measurements, recorded by Beijing's Environment Protection Bureau (EPB), are sparsely distributed (Chen et al. 2015; Cheng et al. 2018). This, coupled with the sharp pollutant concentration gradients that exist across urban areas (Hood et al. 2018), limits the accuracy of any subsequent human exposure analyses. Therefore, air quality modelling, evaluated using network measurements, may fill in the gaps to provide complete spatially and temporally resolved concentration fields (Bates et al. 2018).

Air quality modelling, from global to street-scale, requires detailed representations of local and regional emission fields. However, generating accurate and up-to-date emissions data is a considerable challenge, owing to difficulties in obtaining the necessary activity, emission factor, and production/control technology data for each emission source sector (including power, industry, transport, residential or agriculture) (Hong et al. 2017; Qi et al. 2017). Additionally, in China, the rapid decrease in emissions of major air pollutants over recent years needs to be accounted for (Zheng et al. 2018). This reduction in emissions has followed the nationwide implementation of a number of clean air policies since 2013 as part of the Air Pollution Prevention and Control Action Plan (APPCAP) and more locally through the Beijing Action Plan (Cheng et al. 2018). Overall, emissions in Beijing of $SO_2$, $NO_x$, VOCs and $PM_{2.5}$ are reported to have reduced by 84, 43, 42 and 55% between 2013 and 2017 (Cheng et al. 2018). These emission reductions were estimated by Cheng et al. (2018), using the technology-based model framework of the Multi-resolution Emission Inventory for China (MEIC), and are in good agreement with independent satellite-derived emission trends (Liu et al. 2016; Liu et al. 2017).

The MEIC emission inventory is widely used in studies aimed at understanding the key emission sources and the effectiveness of air pollution control measures across various regions of China (Li et al. 2017; Cheng et al. 2018; Zheng et al. 2018). However, uncertainties in MEIC emissions estimates, related to its underlying methodology and input data, have also been highlighted. For instance, the MEIC model relies on the use of national and provincial energy consumption statistics, which were shown by Hong et al. (2017) to contain large sources of error. The MEIC model uses spatial proxies, such as Gross Domestic Product (GDP) and urban population density, to downscale emissions from provincial to county and grid level scale (Qi et al. 2017). A study by Zheng et al. (2017) revealed a tendency to over-allocate emissions to central urban areas when using these spatial proxies to produce the MEIC inventory at resolutions finer than 0.25°. Zheng et al. (2017) attributed this to the displacement of large industrial facilities away from urban centres, therefore decoupling the real-world location of the emissions from the population-related proxies used to represent them in the MEIC inventory.

Numerous regional modelling studies have been performed for Beijing. Eulerian chemical transport models (CTMs) such as the Nested Air Quality Prediction and Modeling System (NAQPMS), the Comprehensive Air Quality Model with Extensions (CAMx), and the joint Weather Research and Forecast - Community Multi-scale Air Quality models (WRF-CMAQ), have been used to investigate the relative contributions of local and regional emissions to $PM_{2.5}$ concentrations during severe haze periods (Li et al. 2017; Wang et al. 2017; Chang et al. 2019). The WRF-CMAQ model was used by Cheng



et al. (2018) to deduce the relative roles of local and regional emission controls, and favourable meteorological conditions in reducing PM$_{2.5}$ concentrations in Beijing. Further CTM applications have included the assessment of different emission inventory performance as well as the investigation of aerosol-radiative feedback impacts on boundary layer stability and the rate of PM$_{2.5}$ accumulation during heavy haze events (Liu et al. 2016; Petaja et al. 2016; Wang et al. 2018).

A key limitation of regional models, however, is that they cannot be used to represent pollutant concentrations at the
scale needed to fully assess human health impacts. Hence, Gaussian plume dispersion models, capable of simulating pollutant dispersion from explicitly represented emission sources, are often implemented. Widely used for environmental regulatory purposes, models such as ADMS-Urban (Owen et al. 2000) and AERMOD (Cimorelli et al. 2005) incorporate detailed transport processes and simplified fast chemistry schemes, commonly including NO$_x$-O$_3$ reactions, enabling the sharp concentration gradients adjacent to major urban sources to be captured (Hood et al. 2018). Modelling studies, with ADMS-
Urban, have been performed at various degrees of complexity, in terms of emission sources represented, in a number of cities globally. For example, Munir and Habeebullah (2018) investigated roadside PM$_{2.5}$ and PM$_{10}$ levels in Makkah, Saudi Arabia, using only a bottom-up road traffic emissions inventory and Dédelé and Miskinyte (2015) simulated NO$_2$ concentrations in Kaunas City, Lithuania, using transport, industrial and residential sector emissions but excluded buildings information. Hood et al. (2018) incorporated traffic emissions adjustments to account for differences between real-world and test driving cycles,
as well as comprehensive 3-D buildings data, to evaluate the performance of ADMS-Urban across London, UK.

In China, applications of ADMS-Urban have focussed on evaluating the impact that emission control schemes targeting individual sources have on the immediate environment. For instance, Chen et al. (2009) combined pollutant concentrations simulated by ADMS-Urban with population data to investigate the impact of traffic control policies on human exposure levels in Shanghai. Similarly, Cai and Xie (2011) used the ADMS-Urban model to quantify the effect that the odd-
even traffic scheme (restricting vehicles with odd or even number plates), enforced during the 2008 Olympics, had on emissions from a selection of major roads, finding that some of the previously most polluted areas subsequently complied with the Chinese National Air Quality Standards (CNAAQS).

This study aims to produce, for the first time, fully resolved street-scale NO$_2$, O$_3$ and PM$_{2.5}$ concentrations across urban and suburban Beijing using ADMS-Urban and explicit source road traffic emissions. Compiling the traffic emissions
inventory involves apportioning gridded emissions onto the freely available OpenStreetMap (OSM) road network geometry. This approach is much less data-intensive than alternative vehicle activity-based bottom-up methodologies such as the link-level traffic emissions inventory developed for Beijing by Yang et al. (2019) using congestion maps and manual vehicle counts. Both the MEIC v1.3 and an optimised version of the same inventory are used to assess the performance of proxy-based inventories for street-scale modelling. Aggregated sectoral emissions (industrial, power and residential) are also included.
Simulations are performed for the Atmospheric Pollution and Human Health in a Chinese megacity (APHH-China) winter measurement campaign period, which took place in November-December 2016 at the Institute of Atmospheric Physics (IAP), Chinese Academy of Sciences (Shi et al. 2019).

A detailed description of the ADMS-Urban model and its inputs is provided in Sect. 2. Section 3 presents an evaluation and discussion of results comparing modelled concentrations, using both emission inventories, with monitoring network and
field campaign measurement data. Sensitivity studies are included to explore the impact that explicit road traffic emissions, modified diurnal emissions profiles and Beijing's evening urban heat island (UHI) have on measured and simulated pollutant concentration agreement. A summary of this work's primary findings is provided in Sect. 4 along with details of possible future study development.





## 2 Methodology

The street-scale air pollution dispersion and chemistry model, ADMS-Urban, is used here to simulate ambient concentrations of $NO_x$, $NO_2$, $O_3$ and $PM_{2.5}$ across Beijing during the APHH-China winter campaign period (5 November 2016 – 10 December 2016). Section 2.1 provides a full description of the model and its configuration for Beijing, including details on emission source types, pollution dispersion, chemical processes and background pollutant concentrations. Emission inventory development , including the construction of an explicit network of road source emissions, is outlined in Sect. 2.2. In Section 130 2.3, the statistical measures used to evaluate model performance are described.

### 2.1 Model Description and Set-up

ADMS-Urban, developed by Cambridge Environmental Research Consultants (CERC), is a quasi-Gaussian pollution dispersion and chemistry model that has been applied worldwide for environmental regulation (McHugh et al. 2005), investigation and assessment of emission control strategies (Cai and Xie, 2011) and generation of high spatial resolution air 135 quality forecasts (Carruthers, 2009).

The model domain (75 km x 90 km) covers urban Beijing, defined here as everywhere within the Sixth Ring Road (marked in Fig. 1), and extends into the suburban counties of Shunyi and Changping to the north and Tongzhou, Daxing and Fangshan to the south, as illustrated by Fig. 1.

### 2.1.1 Emission sources, meteorological inputs and surface parameters

In the model, pollutant emissions are represented as individual plumes dispersing from a range of explicitly represented sources, including point, road, area and volume sources. Aggregate grid sources (2-D and 3-D) are used to account for additional, poorly-defined diffuse emissions (e.g domestic heating or minor roads) (Mohan et al. 2011; Dédelé and Miskinyté, 2015; Hood et al. 2018). Plume dispersion calculations are driven by a single set of meteorological measurements that are representative of upwind conditions and assumed to be homogeneous across the study domain. For this study, we use hourly wind speed and 145 direction, air temperature and cloud cover data from the Beijing Capital International Airport Meteorology Observatory, which is located ~20 km northeast of the Fourth Ring Road (Fig. 1). The input meteorology is processed by the model to calculate parameters that determine the stability and height of the planetary boundary layer (PBL). Cloud cover measurements, along with the time of day and day of year, are used to calculate incoming solar radiation which generates surface sensible heat flux ($F_{\theta0}$), friction velocity ($U_*$) and Monin-Obukhov length ($L_{MO}$) terms via the surface energy balance. $L_{MO}$ is a measure of the 150 relative importance of mechanical turbulence and buoyancy in the PBL and along with surface heat flux terms determines PBL height (PBLH) in the model. Alternatively, measurements of PBLH can be used if available. For this study, simulations are performed using directly input observations of mixed layer height (MLH) derived from ceilometer measurements taken at the IAP field site during the APHH-China campaign (Kotthaus et al. 2016; Shi et al. 2019). The MLH represents the height of the lowest atmospheric layer always in direct contact with the earth's surface resulting from turbulent exchange (Kotthaus and 155 Grimmond, 2018) and is assumed here to equate to the model's PBLH output.

ADMS-Urban calculates the ratio of PBLH to $L_{MO}$, a measure of the relative importance of mechanical turbulence and buoyancy, to generate a continuous PBL stability profile that varies with height. This PBLH/$L_{MO}$ parameterisation controls subsequently generated vertical and horizontal plume spread extents. In unstable conditions, an additional convectively driven turbulence component is calculated. This produces a skewed, non-Gaussian concentration distribution, meaning that for 160 elevated sources the height of maximum concentration and mean height of the plume itself will descend and ascend, respectively (CERC, 2017).

Differences between conditions at the exposed airport meteorological site and the predominantly built-up modelling domain, largely caused by frictional effects of buildings and street canyons that perturb near-surface dynamics locally, are





accounted for through distinct definitions of surface roughness ($Z_0$) and minimum Monin-Obukhov length ($L_{MO}$) in both
environments. $Z_0$ and minimum $L_{MO}$ values of 0.5 m and 30 m, respectively, represent conditions at the meteorological
measurement site. However, greater $Z_0$ and minimum $L_{MO}$ values of 1.5 m and 100 m, respectively, typical of urban areas
dominated by densely packed tall buildings and concrete surfaces (Stewart and Oke, 2012), are used across the modelling
domain and displace the upwind vertical wind speed, wind direction and turbulence profiles derived from the meteorological
measurements.

### 2.1.2    PBL stability adjustment

Both $Z_0$ and minimum $L_{MO}$ definitions prevent the modelled boundary layer from becoming unrealistically stable in urban
areas where the surface radiation balance is perturbed by a number of factors including anthropogenic heat release, building
geometry and the thermal properties of concrete surfaces (Oke, 1982). The resulting positive temperature differential between
urban areas and the surrounding rural environment is referred to as the urban heat island (UHI) effect (Hamilton et al. 2014).
This phenomenon is strongest in the late afternoon and early evening hours, when anthropogenic heat from rush hour traffic
and residential heating systems, as well as incoming solar radiation stored in the urban fabric throughout the day, is released
into a stabilising PBL (Liu et al. 2007).

   For this study, a further restriction on PBL stability has been applied to more comprehensively account for Beijing's
strong evening UHI (Wang et al. 2017). Figure 2 shows how the PBL stability, represented by PBLH/$L_{MO}$, varies diurnally for
the campaign period. The $L_{MO}$ values are derived from a prior model simulation without stability modifications and the
observed MLH (Sect. 2.2.1) is used as the PBLH. PBLH/$L_{MO}$ > 1, -0.3 ≤ PBLH/$L_{MO}$ ≤ 1 and PBLH/$L_{MO}$ < -0.3 represent stable,
neutral and unstable conditions, respectively.

   During the day, the surface net radiation is partitioned between upwards fluxes of sensible and latent heat and the
downwards flux of heat into the ground (Oke, 1982). The version of ADMS-Urban used here (v 4.2) assumes that this ground
heat flux is a constant proportion of the net radiation. In reality, this proportion varies diurnally, peaking around midday when
a greater proportion of incoming solar radiation is stored by the urban fabric (Anandakumar, 1999; Grimmond and Oke, 1999).
The release of this excess heat in the early evening sustains convection in the PBL, prolonging its instability. To account for
this, $L_{MO}$ values have been adjusted from 4-7 pm (PBLH unchanged) producing the modified campaign period mean
PBLH/$L_{MO}$ diurnal profile illustrated in Fig. 2. This adjustment increases sensible and latent heat fluxes, therefore enhancing
the turbulent mixing of air during this early evening period. The 4-7 pm time window is chosen as it coincides with sunset in
November-December in Beijing and it is in agreement with the extended duration of evening sensible heat flux decay in urban
areas, compared with surrounding rural areas, observed by other UHI-related studies (Zhou et al. 2013; Barlow et al. 2015).
Without this adjustment, the model tends to predict overly stable meteorological conditions in the early evening, which can
lead to the over-prediction of pollutant concentrations. It is important to note that the modelled surface heat flux and $L_{MO}$ terms
are calculated independently of the PBLH, so that small positive $L_{MO}$ values can generate an overly stable boundary layer even
when paired with the measured MLH assumed here to represent real-world stability conditions.

### 2.1.3    Chemistry

The chemical transformation of pollutants contained within each dispersing plume is represented using the Generic Reaction
Set (GRS) chemistry scheme (Malkin et al. 2016). Typically, regional CTMs such as WRF-Chem and CMAQ use detailed
chemical mechanisms containing hundreds, or even thousands of reactions involving NO, $NO_2$, $O_3$ and VOCs, including
homogeneous and heterogeneous aerosol production (Sarwar and Luecken, 2008). The GRS, however, simplifies these to the
following seven reactions:





$$ROC + hv \rightarrow RP + ROC \qquad (R1)$$

$$RP + NO \rightarrow NO_2 \qquad (R2)$$

$$NO_2 + hv \rightarrow NO + O_3 \qquad (R3)$$

$$NO + O_3 \rightarrow NO_2 \qquad (R4)$$

$$RP + RP \rightarrow RP \qquad (R5)$$

$$RP + NO_2 \rightarrow SGN \qquad (R6)$$

$$RP + NO_2 \rightarrow SNGN \qquad (R7)$$

where ROC represents Reactive Organic Compounds, RP is the Radical Pool, SGN is the Stable Gaseous Nitrogen product and SNGN is the Stable Non-Gaseous Nitrogen product (CERC, 2017). The inclusion of fast $NO_x$-$O_3$ chemistry, whereby at high $NO_x$ levels, NO consumes $O_3$ (R3 and R4), enables the sharp pollutant species concentration increases, with proximity to
major road or large point sources, to be captured. R1 summarises all of the oxidation and photolysis reactions that lead to radical production from VOCs (Malkin et al. 2016), while R2 and R5 represent subsequent radical loss.

An additional set of reactions involves the production of ammonium sulphate, following the oxidation of $SO_2$ and reactions with water and ammonia, and this provides a source of both $PM_{10}$ and $PM_{2.5}$. Other secondary organic and inorganic components of particulate matter, which can comprise up to a combined 70% of total $PM_{2.5}$ mass in Beijing (Ma *et al.* 2014;
Tao *et al.* 2017; Wang *et al.* 2017), are accounted for in the background concentration field described in Sect. 2.1.4.

### 2.1.4    Background pollutant concentrations

Background pollutant concentrations represent the regional pollution levels on which the local emissions build. For this study, background levels for $NO_2$, $O_3$, $PM_{2.5}$, $PM_{10}$, $SO_2$ and CO are derived directly from hourly air quality measurement data provided by the China National Environmental Monitoring Center (CNEMC) and are assumed to be uniform across the study
domain. Measured concentrations at 12 of the 35 monitoring stations comprising Beijing's administrative air pollution monitoring network, the IAP field site and an additional site 60 km SE of Beijing, situated in the built-up Guangyang district of Langfang in Hebei province, are used to estimate this background concentration field. The locations of these 14 sites are given in Table 1.

For particulate matter ($PM_{2.5}$ and $PM_{10}$), an hourly upwind background concentration is derived using three sites located
to the NW, NE and SE of urban Beijing, with concentrations selected based on wind direction. Particulates have near-surface lifetimes of days to weeks, therefore concentrations in Beijing are heavily influenced by long-range transport (LRT) of both primary and secondary components originating in neighbouring industrial regions (Wang et al. 2017; Cheng et al. 2018). The measured upwind concentration is expected to capture this transported background regional air. Gaseous species such as $NO_2$, have a much shorter lifetime (~ 1 day) and therefore a smaller regional contribution, with concentrations across urban areas
dominated primarily by local traffic sources (Zhang et al. 2014). The $NO_2$ concentrations at the upwind monitoring station were subsequently not deemed representative of the true background value owing to both this greater spatial variation and the proximity of the upwind monitoring stations to local emission sources. Instead, to approximate background values for gaseous species ($NO_2$, $O_3$, CO and $SO_2$), the hourly minimum concentration for each pollutant across the 12 network monitoring stations and the IAP field site is selected, yielding an approximation for the underlying conditions uninfluenced by local sources.



## 2.2 Emissions inventory processing

For this study, ADMS-Urban simulations use both aggregate 3-D grid source and explicit road source emissions of $NO_x$, $NO_2$, $SO_2$, VOC (total), $PM_{2.5}$, $PM_{10}$ and CO derived from a standard and an optimised version of the high-resolution (3 km) MEIC v1.3 emissions inventory. The standard MEIC v1.3 emissions inventory, for 2013, consists of five emission source sectors: transportation, power, industrial, residential and agricultural (Qi et al. 2017). Note that the latter is not used in this study. The transportation sector is estimated following Zheng et al. (2014), in which county-level emissions, derived from county-level vehicle ownership, are downscaled to grids based on road network and road-specific vehicle activity data. Liu et al. (2015) describes the unit-based technique, adopted to generate the power sector emissions, which utilises the Coal-fired Power Plant Emissions Database (CPED), including information on the technologies, activity data, operation situation, emission factors and locations of individual units. The industrial and residential sectors are both calculated with provincial level activity rates and emission factors before downscaling to county and grid level resolutions based on Gross Domestic Product (GDP) and population density distribution (Zheng et al. 2017). To model conditions during the APHH-China winter campaign, the MEIC v1.3 emissions inventory is re-scaled for this study to account for the 2013-2016 emission reductions in Beijing (Sect. 1). According to Cheng et al. (2018), total emissions of $NO_x$ (and $NO_2$), $SO_2$, VOCs and PM ($PM_{2.5}$ and $PM_{10}$) in Beijing were estimated to reduce by 30, 63, 27, 35 and 30%, respectively, between 2013 and 2016. This adjusted MEIC v1.3 emissions inventory is hereafter referred to as MEIC Std.

An alternate optimised version of MEIC v1.3 (hereafter referred to as MEIC Opt) was created (Li et al. 2018), for November and December 2016, with the aim of addressing the over-allocation of emissions to urban areas that occurs when downscaling MEIC v1.3 to fine scales based on proxy data (Zheng et al. 2017). This MEIC Opt inventory was created using the Nested Air Quality Prediction Modeling System (NAQPMS) to perform iterative minimisation of a cost function comparing NAQPMS simulations with winter campaign observations (Li et al. 2018). This optimisation algorithm was used to redistribute MEIC emissions from central urban Beijing to suburban and rural areas, and to adjust their magnitude to represent the campaign period. Both MEIC Std and MEIC Opt inventories comprise of monthly varying emissions with distinct diurnal weighting profiles applied to each emission sector.

Aggregate 3-D grid sources contain the sum of all MEIC emission source sectors (residential, transportation, industrial and power) and consist of seven vertical layers (38, 90, 152, 228, 337, 480 and 660 m). In the absence of sufficient information required to model point source emissions (e.g large power plants) explicitly, ADMS-Urban's 3-D grid sources enable plume release and dispersion from each of the seven grid source heights, accounting for tall emission sources included within the MEIC v1.3 power or industrial sector grids. MEIC Std and MEIC Opt campaign period mean $NO_2$, $NO_x$, $PM_{2.5}$, $PM_{10}$, $SO_2$ and VOC emission rates from 3-D grid sources, aggregated across all, urban and suburban grid cells, are shown in Table 2.

An explicit network of road emissions for Beijing has been constructed based on the MEIC transportation sector emissions. Figure 3 illustrates the pseudo-top down approach adopted here in the absence of detailed information on traffic activity and fleet composition. Figure 3a shows the spatial distribution of the November and December mean MEIC Std transportation sector surface $NO_2$ emissions. The transportation sector emissions of all pollutants are apportioned to individual road segments on a grid cell-by-grid cell basis, using the geographic information system software ArcGIS. The spatial road network of Beijing, presented in Fig. 3b, is provided by the OpenStreetMap dataset (https://openstreetmap.org/) and includes individual road segment type and geometry information. Emissions are mapped onto the road network based on each road segment length and an emissions weighting factor, producing the distribution shown in Fig. 3c, following Eq. (1):

$$Emis_{i,j,k} = \frac{l_{i,j}.w_k}{\sum_{i=1}^{n}(l_{i,j}.w_k)} \times E_j \quad (1)$$



where $l_{i,j}$ represents the length of road segment $i$ in grid cell $j$ of road type $k$. The weighting factor of road type $k$ is given by

$w_k$. $E_j$ and $n$ denote the total traffic emissions and number of road segments in grid cell $j$, respectively. A weighted mean emission rate, based on road segment length, is calculated along segments traversing multiple grid cells in order to avoid discontinuities.

Weighting factors (Table 3) are estimated using road type and width (based on manual inspection of the most frequent number of lanes for each road type), acting as proxies for traffic activity. The magnitude of weighting factors relative to each

other is important, rather than their absolute values, according to Eq. (1). Minor roads were removed from the network to limit the computational expense of each simulation and are instead aggregated within the 3-D grid sources. This methodology is based on the assumption that traffic volume, speed and behaviour are constant across all road type classes listed in Table 3. However, substantial variations in traffic flow characteristics on roads of the same classification within Beijing's urban area have been observed. For example, Jing et al. (2016) used GPS-fitted buses and taxis to collect near real-time traffic data along

the major road types in Beijing, finding much greater levels of congestion closer to the urban centre, causing increased traffic volume and vehicle speed variations. Given that the same emission weighting factors for roads of the same class are applied across the domain and the lack of traffic flow variations on specific roads within cities in the MEIC framework (Zheng et al. 2014), the methodology adopted here may under-allocate emissions on more congested inner-city roads and over-allocate emissions in suburban areas.

**2.3    Model evaluation**

Evaluation of regional-scale Eulerian CTMs involves the comparison of measurements at specific monitoring site locations with simulated concentrations in the nearest model grid box (Zhong et al. 2016; Wang et al. 2017; Zheng et al. 2017). However, for street-scale air quality modelling with ADMS-Urban, pollutant concentrations can be simulated at specific locations, referred to hereafter as receptor points. For this study, concentrations are modelled at the locations of the 12 monitoring

network stations, as well as the IAP field site, for direct comparison with the corresponding measured concentrations. The following three statistical performance measures are considered simultaneously enabling a comprehensive evaluation of modelled predictions of concentrations, using both MEIC Std and MEIC Opt emissions inventories, during the APHH-China winter campaign period:

$$Normalised\ mean\ square\ error\ (NMSE) = \frac{\overline{(M-O)^2}}{\overline{MO}} \tag{2}$$


$$Fb\ (Fractional\ bias) = \frac{\overline{M} - \overline{O}}{0.5(\overline{O} + \overline{M})} \tag{3}$$

$$Pearson's\ correlation\ coefficient\ (R) \tag{4}$$
$$= \frac{1}{n-1}\sum_{i=1}^{n}(M_i - \overline{M}/\sigma_M)\ (O_i - \overline{O}/\sigma_O)$$

where $n$ denotes the total number of matching hourly modelled and observed concentrations; $\overline{M}$ and $\overline{O}$ indicate mean modelled and observed concentrations, respectively, and $\sigma$ is the standard deviation.

NMSE (ideal value = 0) is a measure of the model's overall accuracy (Cai and Xie, 2011), incorporating the effects of both systematic and random errors (Patryl and Galeriu, 2011); Fb (ideal value =0) reflects the model's tendency to overestimate or underestimate concentrations, compared to measurements; and R (ideal value =1) informs on the extent to which modelled

and measured values are linearly related.





## 3    Results and Discussion

Street-scale resolution maps of PM$_{2.5}$, NO$_2$ and O$_3$ concentrations across a region of urban Beijing are presented in Sect. 3.1. Section 3.2 provides a statistical evaluation of simulated pollutant species against hourly measurements at 12 monitoring network sites and the IAP campaign field site (Table 1), using both MEIC Std and MEIC Opt inventories. Diurnal cycles of

NO$_x$, NO$_2$ and O$_3$ concentrations are given in Sect. 3.3, and Sect. 3.4 contains an analysis of local and regional PM$_{2.5}$ sources. Sensitivity studies explore the impact on model performance of including explicit road emission sources, varying diurnal emissions profiles and accounting for the evening UHI in Sect. 3.5, 3.6 and 3.7, respectively.

### 3.1        Street-scale variation of PM$_{2.5}$, NO$_2$ and O$_3$ concentrations

Mean PM$_{2.5}$, NO$_2$ and O$_3$ concentrations simulated for the campaign period (5 November-10 December 2016), using the MEIC

Opt inventory, for a region of urban Beijing within the Fifth Ring Road are presented in Fig. 4. The influence of the explicit road emissions network on the spatial variation of all species is clear, most notably along the Second, Third and Fourth Ring Roads. PM$_{2.5}$ and NO$_2$ concentrations peak at 125 µg m$^{-3}$ and 160 µg m$^{-3}$, respectively, along the ring road centrelines, before sharply decaying. The magnitude of this drop and distance across which it occurs is determined not only by emission source strength but also by physical and chemical mechanisms, with the speed of plume dispersion and mixing, controlled by

mechanical and convective turbulence generation, interacting with the differing lifetimes of individual pollutants. In Fig. 5, mean NO$_2$ concentrations reduce by ~20-25 µg m$^{-3}$ along a horizontal profile extending 100 m either side of the Second Ring Road. The spatial variation of O$_3$ concentrations is approximately inversely related to these NO$_2$ levels. Modelled O$_3$ concentrations reduce to 5 µg m$^{-3}$ along the Second Ring Road centreline and reach 25 µg m$^{-3}$ between the Fourth and Fifth Ring Roads (Fig. 4). This is a result of the fast reaction of O$_3$ with NO (titration) (R4) which dominates in high NO$_x$

environments (Zhang et al. 2015; Tang et al. 2017; Ma et al. 2018), such as those next to major roads. The conversion of primary NO exhaust emissions to NO$_2$, following the titration of O$_3$, also produces a sharply increasing NO$_2$/NO$_x$ ratio with distance from road centre (Fig. 5). In the following sections, a comprehensive evaluation of model performance is presented.

### 3.2        Model evaluation and assessment of emission inventories

Table 4 summarises the performance of ADMS-Urban in Beijing during the APHH-China winter measurement campaign, with

comparisons between MEIC Std and MEIC Opt simulations enabling an assessment of the MEIC v1.3 optimisation.

Modelled NO$_x$ concentrations at the IAP field site display the most substantial differences between the two simulations (Table 4; Fig. 6). Modelled NO$_x$ concentrations using the MEIC Opt inventory are 149.8 µg m$^{-3}$, 56% lower than the MEIC Std case, leading to NMSE and Fb decreases from 2.35 to 0.63 and 0.93 to 0.17, respectively (Table 4). This enhanced model agreement is reflected in Fig. 6, in which a large proportion of modelled NO$_x$ values reaching 400-600 µg m$^{-3}$

with MEIC Std, up to a factor of six higher than measurements, are reduced to within a factor of two of measured concentrations using MEIC Opt. This result reflects the 63% NO$_x$ emissions reduction across urban Beijing, over all source sectors, in the optimised inventory (Table 2). However, correlation coefficient (R) values for simulated NO$_x$ remain low using both emissions inventories, slightly increasing from 0.35 to 0.41 with MEIC Opt. This smaller improvement in the correlation between measured and modelled NO$_x$ using MEIC Opt, compared to NMSE and Fb, reflects the dependency of R on modelled

NO$_x$ levels that capture the correct temporal variation as well as the overall magnitude of NO$_x$ measurements. The noise apparent in the measured and simulated NO$_x$ comparison in Fig. 6 is therefore likely related to either the diurnal emissions profile or meteorological variations.

NO$_2$ concentrations differ less, with NMSE values of 0.27 and 0.30 for the MEIC Std and MEIC Opt simulations, respectively. However, a greater difference is evident at urban receptor locations, with modelled NO$_2$ concentrations from the

MEIC Opt simulation 12% lower than those with MEIC Std. Across suburban sites, the opposite pattern is seen, with changes



in Fb values between measurements and MEIC Std and MEIC Opt simulations ranging from negative (-0.08) to positive (0.02), respectively.  Both urban and suburban $NO_2$ concentration changes, between simulations, reflect the overall redistribution of $NO_2$ emissions in the MEIC Opt inventory, away from central Beijing and towards the city outskirts (Table 2).

The much greater urban $NO_x$ concentration difference between the two simulations, as compared to $NO_2$, can be attributed to two factors. Firstly, $NO_2$ concentrations respond in a more non-linear way to $NO_x$ emission changes than $NO_x$ concentrations. This has been shown in previous studies (e.g Kurtenbach et al. 2012) and can be explained by the timescales and kinetics involved in the formation and destruction of secondary $NO_2$. As $NO_x$ levels decrease, the production of secondary $NO_2$ via R4 occurs faster as $O_3$ concentrations are higher, leading to a slower rate of decrease of $NO_2$ concentrations compared to $NO_x$ emissions. Additionally, the proportion of $NO_x$ directly emitted as $NO_2$ is greater with MEIC Opt ($NO_2/NO_x$ = 0.093)

than MEIC Std ($NO_2/NO_x$ = 0.068) (Table 2). This is reflected by the much greater reduction, from MEIC Std to MEIC Opt, in domain-aggregated $NO_x$ emissions (43%), as compared $NO_2$ (22%) (Table 2).

At urban sites, $O_3$ concentrations simulated with MEIC Opt are 12.8 µg m$^{-3}$, which is a factor of two greater than those simulated using MEIC Std (6.1 µg m$^{-3}$). Overall, the modelled $O_3$ concentrations at urban sites using MEIC Opt are in closer agreement with the measurements, reflected by lower Fb and NMSE values of -0.29 and 0.93, respectively, as compared

to -0.95 and 3.2 in the MEIC Std simulations (Table 4). This is caused by both lower urban $NO_x$ emissions in MEIC Opt and the reduced proportion of remaining $NO_x$ emitted directly as NO, in MEIC Opt, leading to less $O_3$ destruction through R4. Contrastingly, higher MEIC Opt $NO_x$ emissions in suburban Beijing reduce modelled $O_3$ concentrations by 7%. As a result, modelled $O_3$ performance across all monitoring stations is substantially improved in the MEIC Opt simulation, with a NMSE reduction from 1.54 to 0.74 and an R value increase from 0.71 to 0.79 (Table 4). These results highlight the strong dependency

of $O_3$ concentration predictions in urban areas, which inform human exposure analyses and influence future emission control implementation, on the accurate spatial variation and magnitude of $NO_x$ emissions in high resolution emissions inventories. The increase in modelled urban $O_3$ concentrations following $NO_x$ emissions reductions also highlights both the negative impact that controls on one pollutant species can have on another as well as the possible need for air quality guidelines that consider multiple pollutants in contrast with the single pollutant-based air quality index used in China (Han et al. 2018).

Figures 7a and 7b illustrate site-specific differences between measured and simulated $NO_2$ and $O_3$ concentrations, respectively, using both emissions inventories. It is clear that, despite generally closer model agreement with measurements using MEIC Opt, $NO_2$ concentrations remain substantially overestimated at urban sites 1 (~14 µg m$^{-3}$) and 2 (~9 µg m$^{-3}$). To help understand the cause of this, the sensitivity of modelled concentrations within 100 m of a road source near site 1 is illustrated in Fig. 8. Concentrations along a cross-road slice, extending 100 m either side of the road, are simulated after halving

and doubling the magnitude of emissions of all species from this secondary road. Emissions from all other sources in the model configuration remain the same. Along the road centre, the range of simulated concentrations between emission scenarios is ~10 µg m$^{-3}$, however this difference decreases to ~2 µg m$^{-3}$ at a distance of 100 m, which is much lower than modelled $NO_2$ overestimations produced by MEIC Opt at sites 1 and 2, each located ~80-90 m from the nearest road . Therefore, the high modelled $NO_2$ at sites 1 and 2 may only be partially attributed to an over-allocation of explicit road source emissions caused

by either (a) underlying gridded emissions that are still too high, or (b) not considering traffic volume/speed variations across the domain in road class emission weighting factor estimates. It should also be noted that the physical barriers to pollution dispersion represented by the urban canopy, and specifically street canyons, are not explicitly modelled in this study. This may lead to road emissions dispersing further from the road centre than in reality, therefore contributing to elevated modelled concentrations at greater distances from explicit roads sources (Dédélé and Miskinyte, 2015). The sensitivity of the simulated

$NO_2/NO_x$ concentration ratio to emission magnitude changes is also shown in Fig. 8. For each emissions scenario, the $NO_2/NO_x$ emission ratio remains the same (0.093) (Table 2), however the concentration ratio varies. With doubled $NO_x$ emissions, the $NO_2/NO_x$ ratio is ~0.3 along the road centre, compared to ~0.4 with halved $NO_x$ emissions (Fig. 8). This difference, which decays to zero at a distance of ~75 m, is mostly driven by PBL dynamics and the mixing of freshly emitted $NO_x$ into air with



a lower $NO_2/NO_x$ concentration ratio driven by the impact of higher $NO_x$ emissions on secondary $NO_2$ production via R4, as discussed above.

A clear distinction exists between measured $PM_{2.5}$ concentrations recorded at the suburban (78 µg m$^{-3}$) and urban (101 µg m$^{-3}$) monitoring stations (Table 4). These values are well in excess of China's annual $PM_{2.5}$ National Ambient Air Quality Standard (NAAQS) of 35 µg m$^{-3}$, however concentrations are expected to be higher in winter due to more stable meteorology (Zheng et al. 2015; Li et al. 2017) and enhanced coal combustion for residential heating and cooking and at power plants in neighbouring cities (Chen et al. 2017). Simulated $PM_{2.5}$ concentrations, however, do not reflect such an urban-suburban discrepancy, with mean urban values exceeding those at suburban sites by only 9 µg m$^{-3}$ and 7 µg m$^{-3}$ using MEIC Std and MEIC Opt, respectively (Table 4). Across all monitoring stations, the range in campaign period mean measured concentrations is substantially higher (~40 µg m$^{-3}$) than the simulated range using both MEIC Std (~20 µg m$^{-3}$) and MEIC Opt (~15 µg m$^{-3}$), respectively. These results suggest that either $PM_{2.5}$ emission sources are too uniform in magnitude and spatial distribution across the domain in the current model set-up, or that the assumption of a homogeneous background $PM_{2.5}$ concentration is invalid. It is likely that, by diluting $PM_{2.5}$ emissions within individual grid cells and not explicitly representing point source emissions (e.g. large industrial units), the model is unable to capture the $PM_{2.5}$ concentration hotspots that would increase the urban $PM_{2.5}$ level increment and improve model agreement with the observed spatial variation. With the exception of simulated $PM_{2.5}$ adjacent to major roads, this modelled uniformity in urban $PM_{2.5}$ is clearly evident in Fig. 4, in which $PM_{2.5}$ concentrations vary by only ~5-10 µg m$^{-3}$ across the area enclosed by the Fifth Ring Road. The mean estimated $PM_{2.5}$ background concentration is 79 µg m$^{-3}$ (Fig. 7), which is higher than both the mean measured concentrations at suburban sites 3 and 10, located to the north. This implies that either the background $PM_{2.5}$ level is, in reality, inhomogeneous with lower concentrations to the north and higher to the south of urban Beijing, or that the upwind background monitoring site to the south is too heavily influenced by local emission sources and is not representative of background conditions. The relative contributions from $PM_{2.5}$ emission sources and background inhomogeneity to the underestimated spatial variation in $PM_{2.5}$ is further discussed in Sect. 3.4.

### 3.3 Winter campaign diurnal cycles of $NO_2$, $O_3$ and $NO_x$

The diurnal variation of pollutant species in urban areas provides insight into how concentrations are impacted by both diurnal variations in meteorology and temporally varying emissions. The locations of urban stations 1, 12 and IAP, and suburban site 11 are illustrated in Fig. 9, with measured mean diurnal $NO_2$ concentrations averaged over the campaign period at all four sites compared with those simulated using both the MEIC Std and MEIC Opt inventories in Fig. 10. There are several common differences between modelled and measured concentration profiles at all three urban stations (Fig. 10a, 10c and 10d). Firstly, both simulated $NO_2$ diurnal cycles at sites 1, 12 and IAP are considerably lower than measurements from 11 pm to 6 am. This discrepancy peaks at 2 am, with simulated concentrations ~20 µg m$^{-3}$ and ~30 µg m$^{-3}$ lower than measurements, using MEIC Std and MEIC Opt, respectively. Observed $NO_2$ concentrations at urban sites remain elevated between 11 pm and 6 am, relative to the rest of the day, resulting in a diurnal profile absent of the distinct morning and evening peaks commonly observed in other megacities, such as London (Hood et al. 2018). High nocturnal $NO_2$ concentration measurements during the APHH-China winter campaign at the IAP field site are also noted by Shi et al. (2019).

Previous studies have attributed the evening influx of heavy duty diesel trucks (HDDTs), banned from commuting within the Fourth Ring Road from 6 am to 11 pm (Zhang et al. 2019), to evening $NO_x$ concentration increases across urban Beijing of up to 10 µg m$^{-3}$ (Wu et al. 2016; Yang et al. 2019). A large proportion of this HDDT fleet originates in other provinces where emission standards are not as strict as those in Beijing (Wang et al. 2011). It is possible, therefore, that in a proxy-based emissions inventory (e.g MEIC), such traffic restrictions and inter-provincial vehicle mobility are not fully accounted for (Zheng et al. 2014). This is supported by the much closer agreement between evening modelled and measured $NO_2$ at suburban site 11 (Fig. 10b), situated outside the Sixth Ring Road (Fig. 9) and away from the influence of additional



nighttime HDDT NO$_x$ emissions. Additionally, ADMS-Urban makes an approximation when modelling dispersion in calm conditions by applying a minimum wind speed of 0.75 m s$^{-1}$ (CERC, 2017). These stable, low wind speed conditions, however, are common in winter in Beijing and have been strongly linked to the acceleration of pollution accumulation during severe winter haze events (Zhang et al. 2015; Zhang et al. 2016; Zhang et al. 2018). Therefore, it is likely that the large early morning

NO$_2$ concentration model underestimations across all three urban sites are a consequence of NO$_x$ emissions that are too low in magnitude, from 11 pm to 6 am, dispersing into a simulated PBL that may be insufficiently stable due to the use of a minimum wind speed in the model.

From 6-9 am, modelled NO$_2$ concentrations in both the MEIC Std and MEIC Opt simulations increase sharply (Fig. 10). This is most prominent at site 1, where simulated levels approximately double during this three hour period. This increase

corresponds to the release of rush hour traffic-related NO$_x$ emissions into a stable and shallow morning PBL. Contrastingly, measurements at these sites decline over this early morning period following an evening concentration peak as described above. This overestimation of NO$_2$ continues throughout the afternoon, with similar profiles at sites 12 and IAP reflecting the close proximity of both receptor locations (~3 km apart) (Fig. 9).

The concurrence of evening rush hour traffic emissions and a stabilising PBL, associated with the reduction in surface

heating following sunset, creates a second simulated NO$_2$ concentration peak at ~6 pm. In contrast to the simulated morning concentration rise, the close agreement between the measured and modelled time of onset and magnitude of this early evening increment indicates that the simulated stability adjustment (Sect. 2.1.2), implemented between 4-7 pm, has successfully accounted for the re-release of stored heat, characteristic of large urban areas.

There is little difference between the diurnal NO$_2$ concentration profiles simulated using the MEIC Std and MEIC

Opt inventories and this is consistent with the model evaluation results described in Sect. 3.2. Simulated NO$_2$ concentrations across the urban sites are marginally lower using MEIC Opt compared to MEIC Std, with the reverse true at suburban site 11, again reflecting the relocation of emissions out of the urban centre.

The much closer agreement between measured NO$_x$ concentrations at the IAP field site and those simulated with MEIC Opt compared to MEIC Std, outlined in Sect. 3.2, is further emphasised by the diurnal cycles in Fig. 11. MEIC Opt

produces much lower NO$_x$ concentrations than MEIC Std across all hours of the day (up to a factor of three), peaking during morning and evening rush hour with concentrations of ~200 µg m$^{-3}$ and ~250 µg m$^{-3}$, respectively. The simulated NO$_2$/NO$_x$ concentration ratio at IAP produced with MEIC Opt ranges from 0.4 to 0.7 throughout the day, 0.2-0.3 greater than the MEIC Std simulation. This again reflects the combined influences of both the greater NO$_2$/NO$_x$ emission ratio in MEIC Opt (Table 2) and the non-linear response of secondary NO$_2$ concentrations to NO$_x$ emission changes, as discussed in Sect. 3.2.

Overestimated NO$_x$ concentrations and underestimated NO$_2$/NO$_x$ concentrations ratios at IAP produced with MEIC Opt indicate that, despite emissions modifications, the magnitude of NO$_x$ emissions (specifically NO), are too high in the MEIC Opt inventory.

The impact of NO$_x$ emissions differences on diurnal O$_3$ concentrations is illustrated in Fig. 12. Using MEIC Std, simulated O$_3$ concentrations across all three urban sites, are considerably lower than measured values from 8 am to 5 pm, with the

measured-modelled difference reaching ~20 µg m$^{-3}$ at midday. This reflects the prominence of R4, caused by high urban NO emissions in MEIC Std. The reverse response is seen at site 11, where midday O$_3$ is overestimated by ~10 µg m$^{-3}$ as a result of low MEIC Std NO emissions in suburban versus urban regions. During daylight hours, there is much closer agreement between measured and modelled O$_3$ across all four sites with MEIC Opt. This reflects the adjusted balance between photochemical production of O$_3$, via R3, and its removal via R4, caused by decreased NO$_x$ emissions in urban areas and

increased emissions in suburban areas, in the MEIC Opt inventory. NO$_x$-O3 chemistry is also greatly influenced by proximity to road sources. As shown in Fig. 8 and discussed in Sect. 3.2, roads with higher NO$_x$ emissions lead to lower NO$_2$/NO$_x$ concentration ratios within distances of 100m and therefore greater O$_3$ loss through its titration by NO in R4.



### 3.4 Local and regional contributions to PM$_{2.5}$ concentrations

Figure 13 presents the diurnal variation of the range of site-specific campaign period mean measured and simulated PM$_{2.5}$ concentrations, using MEIC Opt, across all 12 monitoring network sites. The interquartile range of all network measurements, illustrating the extent to which PM$_{2.5}$ concentrations vary spatially across the domain, greatly exceeds that of modelled concentrations for most of the day. This observed range is largest at night and consistently in excess of 20 μg m$^{-3}$, compared to the simulated range of 5-10 μg m$^{-3}$. The measured ranges are additionally sub-divided into those recorded at urban and suburban monitoring sites, with the diurnal median urban PM$_{2.5}$ values as much as ~35 μg m$^{-3}$ higher than those for suburban sites between 11 pm and 2 am. It is possible that, similarly to the elevated evening NO$_2$ concentration measurements discussed in Sect. 3.3, this high measured nighttime urban PM$_{2.5}$ concentration increment is also related to the influx of HDDTs to central Beijing following the lifting of traffic restrictions from 11 pm to 6 am, with recent studies (Zhang et al. 2015; Wu et al. 2016) reporting a rising contribution from HDDT exhaust emissions to PM$_{2.5}$ levels across China. A subsequent reduction of the measured urban-suburban PM$_{2.5}$ level discrepancy during daytime hours, reaching ~10 μg m$^{-3}$ at midday, coincides with much closer overall agreement between modelled and measured concentrations.

The large difference between mean measured urban and suburban PM$_{2.5}$ concentrations throughout the day in Fig. 13 is not captured by the model. This is likely the result of either a lack of heterogeneity in the modelled PM$_{2.5}$ emission sources, particularly across urban areas, or that, in reality, substantial non-uniformity in the background concentration exists across the domain. The former is consistent with a number of previous studies on PM$_{2.5}$ source apportionment in Beijing, which have suggested that, during extended periods of elevated particulate mass concentrations in winter, local emissions can account for 80% of PM$_{2.5}$ concentrations (Li et al. 2017; Wang et al. 2017; Chang et al. 2019). Therefore, as discussed in Sect. 3.2, in order to simulate the high spatial variation of PM$_{2.5}$ concentrations, characterised by large urban PM$_{2.5}$ concentration increments, higher resolution modelling of primary PM$_{2.5}$ emissions, through the inclusion of explicitly represented large point sources is likely to be necessary.

It is also possible that greater secondary aerosol production needs to be included in the model's chemistry scheme, further increasing the simulated urban PM$_{2.5}$ increment. Currently in ADMS-Urban, with the exception of ammonium sulphate production, secondary PM$_{2.5}$ concentrations are assumed to be included in the upwind background concentration. As the dominant contribution to secondary PM$_{2.5}$ in Beijing is reported to be from neighbouring industrial regions to the south (Ma et al. 2017), this assumption seems largely valid. However, the relative local contributions of other secondary components in Beijing, such as ammonium nitrates, are found to be increasing (Wang et al. 2017; Xu et al. 2019; Yang et al. 2019). This is a consequence of the effectiveness of recent SO$_2$ emission controls and the lack of agricultural ammonia (NH$_3$) emissions reductions (Zheng et al. 2018), which have promoted the formation of ammonium nitrate (Xu et al. 2019). Xu et al. (2019) also found the nitrate aerosol to be of increasing importance at night during winter, as a result of its greater stability at lower temperatures, which, coupled with high nighttime NO$_2$ concentrations (Fig. 10), may further account for the elevated evening PM$_{2.5}$ levels (Fig. 13). The applicability of this previous work is possibly limited by the smaller domain size and short timescales of pollution dispersion in this study compared with those necessary for secondary aerosol production. However, future work testing the impact of both the higher resolution representation of PM$_{2.5}$ emission sources and additional secondary particle formation pathways within the chemistry scheme is needed to fully understand the potential impact of both on improving agreement between simulated and measured PM$_{2.5}$ concentrations (Fig. 13).

The regional contribution to total PM$_{2.5}$ concentrations in Beijing has been shown by previous studies to vary from <10% to >90% depending on the time of year and meteorological conditions (He et al. 2015; Liu et al. 2015; Li et al 2017; Wang et al. 2017). Therefore, the sensitivity of the calculated PM$_{2.5}$ background concentration to the methodology used to select the appropriate monitoring site is important and is illustrated in Fig. 14. As described in Sect. 2.1.4, simulated PM$_{2.5}$ concentrations include a wind direction-dependent upwind background contribution calculated using either of two sites to the



north or one to the south of urban Beijing. Figure 14 shows the diurnal range of calculated upwind background values during

the winter campaign, with the corresponding range of background $PM_{2.5}$ calculated by instead selecting the minimum hourly

concentration across the monitoring network, matching the methodology used to determine background concentrations for

gaseous species.

For each hour, the upwind background $PM_{2.5}$ upper quartile, median and lower quartile greatly exceed the corresponding

values when using the minimum background methodology. This discrepancy is greatest for the upper quartile values and peaks

during morning and evening rush hour, reaching ~80 µg m$^{-3}$ at 5 pm (Fig. 14). The lower whiskers, however, denoting the

lowest datum lying within 1.5 times the interquartile range of the lower quartile, are common across both sets of $PM_{2.5}$

background diurnal cycles. Interpretation of these results is assisted by Fig. 15, which presents hourly wind vectors and $PM_{2.5}$

time series measurements throughout the campaign at all three upwind sites as well as urban sites 1 and 2. It is clear that the

highest upwind $PM_{2.5}$ background concentrations occur when values at the additional site to the southeast of urban Beijing

(site 14) are selected during periods of southerly winds (Fig. 15). The lowest background concentrations, therefore, can be

attributed to either of the northerly sites (site 3 and site 10). Northerly winds advect clean air originating over the relatively

unpolluted mountainous regions into urban Beijing (Tie et al. 2015). This switch in wind directions creates a saw-tooth pattern,

with pollution episodes initially consisting of a slow build-up phase, associated with stagnant southerly winds, and culminating

with sharp concentration drops related to the influx of cold northerly air (Li et al. 2017; Wang et al. 2017). A clear example of

this, from 23-27 November, is shown in Fig. 15.

$PM_{2.5}$ concentrations at site 14, situated in the south-eastern corner of the model domain are consistently higher than those

measured at sites located in central Beijing. This monitoring station is located in the built-up Guangyang District of Langfang

in Hebei province and is not in the immediate vicinity of any large point sources. Therefore, this region is possibly more

heavily influenced by regional $PM_{2.5}$ advected from industrial towns and cities to the south. This highlights an important

limitation of our study, which assumes a homogeneous background concentration for each species; this assumption may not

be valid across such a large and complex urban area.

Both local $PM_{2.5}$ emission sources not represented in our study and background inhomogeneity appear to contribute

substantially to differences in the spatial and temporal variation of measured and modelled $PM_{2.5}$ concentrations. However,

the large diurnal variability in measured $PM_{2.5}$ concentration ranges across the domain (Fig. 13), not captured by the model, is

more likely the influence of local emission sources, with longer timescales required for background $PM_{2.5}$ concentration

variability driven by regional transport.

### 3.5  Impact of explicit road source modelling

In this section, we investigate the sensitivity of the simulated $NO_2$ concentrations to the inclusion of explicit road source

emissions. Simulations that use aggregate 3-D grid sources alone are much less computationally expensive than those that also

incorporate explicit road source emissions and allow studies to be performed with ADMS-Urban in urban areas where detailed

road network information is unavailable. In Fig. 16, measured $NO_2$ concentrations averaged across the campaign are compared

with those simulated using 3-D grid and explicit road sources, as well as 3-D grid sources only, derived from the MEIC Opt

emissions inventory. By resolving road traffic emissions into explicitly represented road sources, as opposed to using gridded

emissions only, mean modelled $NO_2$ concentrations across urban stations increase from 62.8 µg m$^{-3}$ to 71.4 µg m$^{-3}$ (Table 5).

The corresponding Fb value improvement, at urban sites, from -0.13 to 0 (Table 5), reflects the negative impact that diluting

road traffic emissions over each 3 x 3 km grid cell has on simulated and observed $NO_2$ level agreement across a region densely

populated by major roads.

More accurate model predictions next to roads can lead to better assessments of human exposure levels to pollutant species

and is evidence of the successful implementation of the top-down approach to estimating explicit road traffic emissions used

in this study. However, it is also clear from Fig. 16 that agreement between modelled and measured $NO_2$ concentrations at





sites 1 and 2 is substantially poorer when using explicit road sources than with the grid source only simulation. The model evaluation statistics for all monitoring sites (Table 5), reflect this with increases in NMSE from 0.28 to 0.3 and decreases in R from 0.59 to 0.53 when modelling road emissions explicitly. As discussed in Sect. 3.2, this highlights the impact that the

assumption of constant traffic activity, high underlying gridded emissions or the absence of street canyon and urban canopy modelling can have on simulated concentrations at certain near-road locations. Little change is seen across suburban areas with the inclusion of explicit road source emissions, reflecting the lower density of roads and more dominant contribution from diffuse emissions with distance away from Beijing's urban centre.

### 3.6 Accounting for additional evening NOx emission source

In this section, the influence of modifying the MEIC diurnal emissions profile, used for all previous simulations, to account for additional sources of nighttime $NO_x$ emissions is examined. As discussed in Sect. 3.3, a likely explanation for the simulated underestimate in nocturnal $NO_x$ and $NO_2$ concentrations is that an additional evening $NO_x$ emission source is not accounted for in the emissions inventories. The timing of these peak $NO_2$ and $NO_x$ measurements, between 11 pm and 6 am, coincides with the influx of HDDTs within Beijing's Fourth Ring Road.

Figure 17 presents the standard MEIC diurnal emissions profile (DP_MEIC), and two alternative profiles, DP_25 and DP_50, constructed by increasing the standard MEIC profile factors between 11 pm and 6 am by ~0.25 and ~0.5, respectively, to account for additional nighttime HDDT emissions. For both modified emissions profiles, in order to retain the same 24-hour emissions total, DP_MEIC is further adjusted between 7 am to 10 pm, by magnitudes that also preserve the timings of the morning and evening emissions peaks associated with rush hour traffic. The weekend emissions profile, characterised by a

delayed morning peak and ~30% reduced total daily traffic emissions, is kept unchanged for all three sensitivity simulations. Campaign period mean diurnal profiles of $NO_2$ concentrations, simulated using the diurnal emissions profiles shown in Fig. 17, are presented in Fig. 18. At suburban site 11, the close agreement between simulated and measured $NO_2$ concentrations using DP_MEIC is strengthened further by increasing the proportion of emissions released at night relative to the daytime. Modelled $NO_2$ level overestimations throughout the morning and afternoon hours at sites 12 and IAP using DP_MEIC are

reduced when applying the two modified emissions profiles. However, at site 1 the application of DP_50 is unable to reduce daytime $NO_2$ concentrations substantially, which is likely related again to the effect of overestimated emissions along the nearest explicit road source (Fig. 8). The evening $NO_2$ concentration is underestimated at sites 1, 12 and IAP, using DP_MEIC, and this is successively reduced by a small amount with DP_25 and DP_50. The remaining evening differences suggest that, although the inclusion of higher nighttime emissions improves agreement, other possible issues exist related to ADMS-urban's

inability to model dispersion at very low wind speeds; inaccurate underlying gridded emissions; the simplified GRS chemistry scheme; the exclusion of street canyon and urban canopy modelling or PBL dynamics.

### 3.7 The influence of boundary layer height and stability on diurnal NO₂ concentrations

In this section, the impact of PBLH and stability on diurnal $NO_2$ concentrations is explored with further sensitivity simulations. The space into which emitted plumes of pollutants can disperse and mix is determined by the PBLH. Figure 19 shows the

difference between measured and modelled PBLHs and their impact on simulated diurnal $NO_2$ concentrations. Differences between the PBLH simulated without evening stability adjustment and the observed PBLH (Fig. 19) are characterised by a daytime overprediction and nighttime underprediction. At 3 pm, the rapidly growing convective PBL peaks at ~1100 m, exceeding the observed heights by ~200 m. This difference between observed and simulated PBLHs could be a result of an overestimation of the solar radiation-driven surface sensible heat flux and mechanically-driven turbulent flux values, which

are the principal parameters impacting the modelled PBLH. Additionally, due to complex cloud physics, detecting the exact limit of vertical mixing is difficult in the presence of low level stratiform clouds, which form frequently in Beijing during




winter, and may further account for low PBLHs derived from ceilometer observations (Kotthaus and Grimmond, 2018). After sunset at 5 pm, the modelled PBLH shrinks to ~200 m, 400 m below the measured height. This sharp transition between an unstable and stable modelled PBL is a consequence of ADMS-urban not accounting for the UHI effect in its surface energy
balance calculations, as described in 2.1.2.

The early evening stability adjustment applied to all previous simulations in this study replicates the effect of the UHI by reducing PBL stability between 4-7 pm. By applying the stability modification, early evening modelled PBLH increases and reaches ~1300 m by 6 pm before sharply decreasing to ~200 m by 8 pm. Note that directly input PBLH measurements are unaffected by changes to $L_{MO}$ and surface heat flux terms. The stability adjustment reduces $NO_2$ concentrations simulated at 4
pm using modelled and measured PBLHs by ~40 $\mu$g m$^{-3}$ and ~15 $\mu$g m$^{-3}$, respectively, greatly improving agreement with measurements. The sharp morning modelled $NO_2$ concentration rise, peaking at 9 am, decreases by ~10 $\mu$g m$^{-3}$ through use of the measured PBLH alone. However, application of a similar PBL stability adjustment, between 7-10 am, would likely reduce the early morning PBLH underestimation and further weaken the modelled $NO_2$ concentration rise associated with the input of rush hour-related $NO_x$ emissions into a morning PBL that is currently too stable and too shallow compared to observations.
The results suggest that although atmospheric stability has a strong impact on $NO_2$ concentrations, the use of observed PBLH instead of modelled heights has little effect. This is most notable at night, when much greater measured nocturnal PBLHs have negligible impact on simulated $NO_2$ levels. This is possibly related to the dominant impact in the model configuration of near-surface traffic emission compared to elevated sources, with dispersion of the latter more likely to be restricted by low PBLHs.

## 4    Conclusions

In this study, street-scale resolution concentrations of $NO_x$, $NO_2$, $O_3$ and $PM_{2.5}$ are simulated for Beijing, using the Gaussian pollution dispersion and chemistry model, ADMS-Urban. Simulations for the APHH-China winter measurement campaign period (5 November 2016 – 10 December 2016), are driven by an explicit source road traffic emissions inventory, developed for this work using a pseudo top-down methodology. This approach, which involves apportioning an underlying high-resolution gridded emissions inventory onto Beijing's spatial road network, provided by OpenStreetMap, may be applied to
investigate the air quality in other cities where detailed bottom-up traffic emissions inventories are unavailable.

Measurements recorded at 12 of Beijing's air quality monitoring network stations and at the Institute of Atmospheric Physics (IAP) field site are compared with simulated pollutant levels generated by the Multi-resolution Emission Inventory for China v1.3 (MEIC Std), at 3 km resolution, and an optimised version of the same inventory (MEIC Opt). MEIC Opt, which is based on campaign measurements, has lower emissions across urban Beijing (within the Sixth Ring Road), and higher
emissions in surrounding suburban areas, resulting in greatly improved agreement between observed and simulated concentrations for all species. Most notably, driven by NO emission changes, simulated mean $NO_x$ concentrations at the IAP site are lower by more than a factor of two using MEIC Opt compared to the MEIC Std inventory. Consequently, modelled urban $O_3$ concentrations increase by 109%, with suburban $O_3$ concentrations decreasing by 7% in simulations performed with MEIC Opt.

The inclusion of explicit road sources allows sharp $NO_2$ concentration gradients adjacent to major roads to be resolved, leading to generally closer agreement between network measurements and simulated concentrations. However, limitations of the model configuration can lead to modelled $NO_2$ levels that are substantially higher than measurements at some near-road (~100 m) sites. These model uncertainties stem from the application of uniform weighting factors to roads of the same classification (thus neglecting traffic activity variations), the assumptions inherent to the underlying gridded inventory, and
exclusion of the physical barriers to pollution dispersion created by street canyons.

Differences in the diurnal variability of measured and simulated $NO_2$ concentrations during the winter campaign period reveal features related to emissions (e.g. local driving restrictions) and the Urban Heat Island (UHI), that air quality



modelling studies over large urban areas should consider. For instance, measured $NO_2$ concentrations at urban monitoring sites situated close to roads can reach nighttime values above 80 µg m$^{-3}$, exceeding both morning and evening rush hour levels. This

pattern is not reproduced in the simulated $NO_x$ concentrations and is consistent with the evening influx of heavy duty diesel trucks (HDDTs), banned from traversing within the Fourth Ring Road between 6 am and 11 pm. The increase in HDDT traffic at night across urban Beijing is therefore an important local emission source that needs to be included in MEIC and other proxy-based emission inventories. Additionally, modifying modelled PBL stability parameters to replicate early evening (4-7 pm) instability driven by the delayed release of heat stored in the urban fabric, improves the diurnal variation in simulated $NO_2$

concentrations. A similar modification may improve morning model predictions, although it would be difficult to use the presence of a UHI to justify this.

     The range in measured $PM_{2.5}$ concentrations across the monitoring network for the campaign period (~40 µg m$^{-3}$) is much higher than the corresponding simulated range using both MEIC Std (~20 µg m$^{-3}$) and MEIC Opt (~15 µg m$^{-3}$). The large difference between measured suburban and urban $PM_{2.5}$ levels is also not captured by the model and may indicate any or

all of the following: (a) $PM_{2.5}$ emissions are too low in magnitude and not represented at sufficiently high resolution, particularly across urban areas, (b) the simplified GRS chemistry scheme needs to be modified to increase contributions from locally produced secondary $PM_{2.5}$, or (c) the assumption of a homogeneous background concentration across complex megacities, such as Beijing, which are heavily influenced by the advection of regional pollution, is not valid.

  Sensitivity studies have shown that using explicit road source emissions; including an additional nighttime emission source;

and accounting for UHI effects, through enhanced early evening instability conditions, can produce closer agreement between simulated and measured $NO_2$ concentrations.

     Street-level modelling, along with the open data sources and methodologies used here, may be applied for future work elsewhere. Quantifying spatio-temporal pollutant distributions at such fine scales is essential for human health exposure-related studies, and for informing choices on the emission controls of specific sectors.


*Author contributions*. MB set up and ran the model with support from JS. MB processed the model outputs with support from JS. This manuscript was written by MB with guidance from JS, RMD, OW and DC. All authors read and improved the manuscript. ZS, JL, FAS, SK and SG provided measurement data. JL, QZ, RW and MH provided and processed the emissions data.


*Acknowledgements*. This work was funded by the UK Natural Environment Research Council (NERC) Industrial studentship scheme with CASE support provided by Cambridge Environmental Research Consultants (CERC) under grant code NE/N0077941/1. We would also like to acknowledge the APHH-China programme funded by NERC under the grant codes NE/N006941/1, NE/N006925/1 and NE/N006976/1.

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





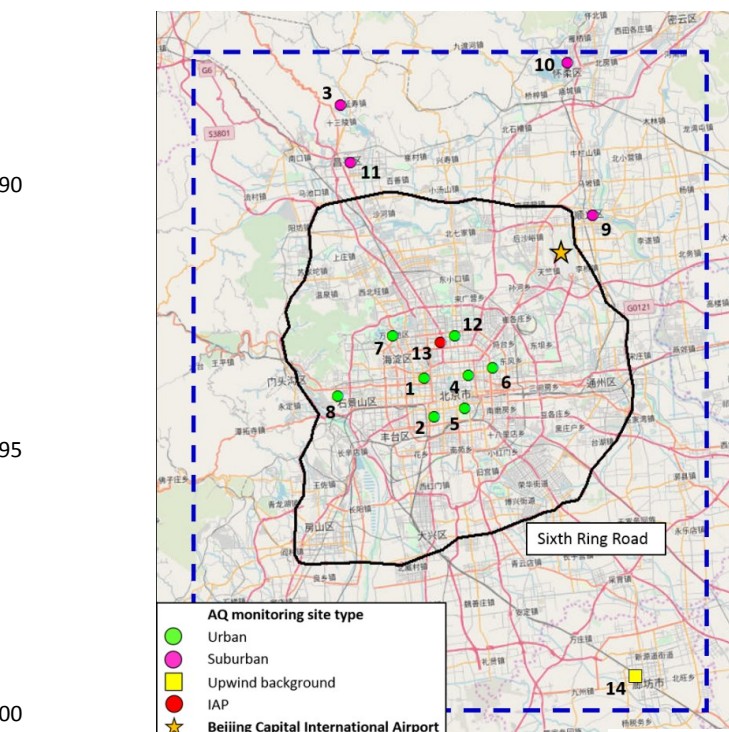

**Figure 1: Map of Beijing (source: OpenStreetMap) with modelling domain, measuring 75 km x 90 km, outlined (dashed blue line). Urban (green circle), suburban (pink circle), upwind background (yellow square) and IAP (red circle) air quality monitoring station locations, including site numbers, are provided. Beijing Capital International Airport Meteorology Observatory (yellow star) and the Sixth Ring Road (black line) are also highlighted. © OpenStreetMap contributors 2019. Distributed under a Creative Commons BY-SA License.**

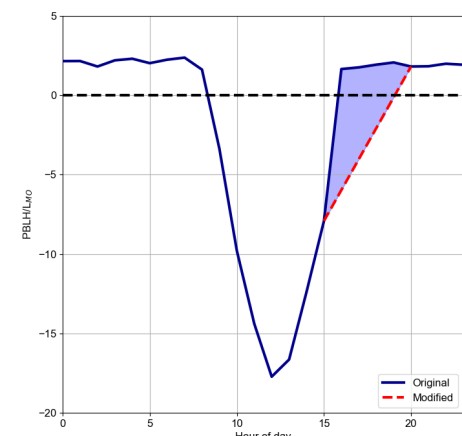

**Figure 2. Diurnal mean $PBLH/L_{MO}$ values for the campaign period (blue line). Modified $PBLH/L_{MO}$, from 4-7pm, to account for evening UHI, shown by red dashed line.**







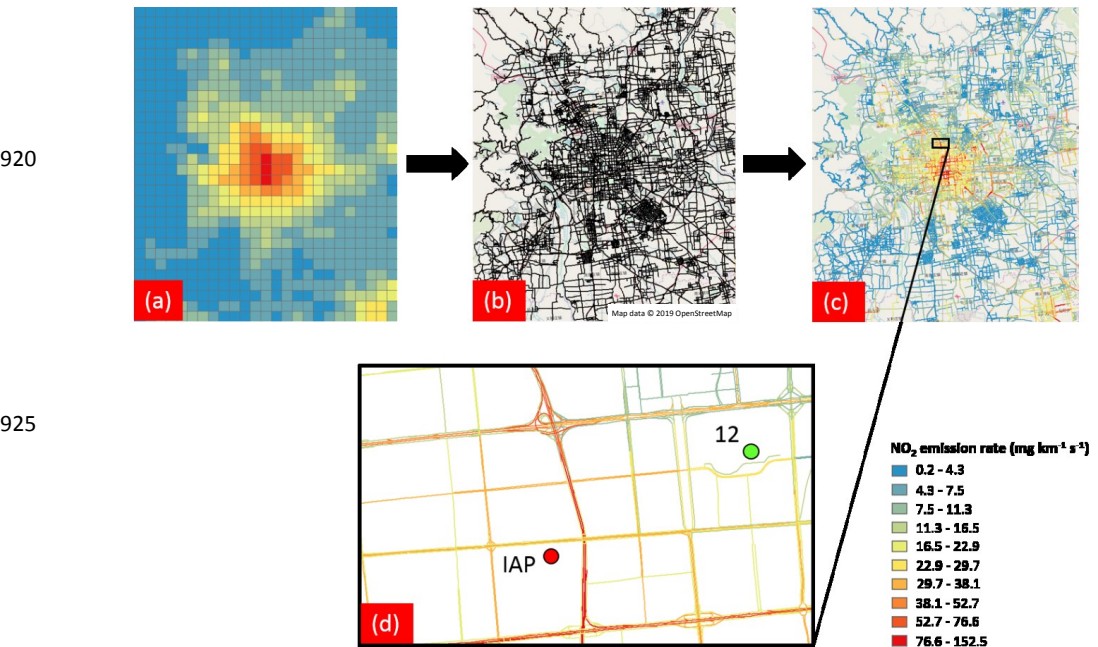


**Figure 3: (a)** Spatial distribution of November and December mean transportation sector MEIC Std $NO_2$ emissions (lowest vertical layer) covering full study domain, **(b)** Spatial road network of Beijing (source: OpenStreetMap, © OpenStreetMap contributors 2019. Distributed under a Creative Commons BY-SA License.), **(c)** explicit road source $NO_2$ emission rates following apportioning of **(a)** onto **(b)**, and **(d)** enlarged Sect. of road emissions network covering the IAP field site and site 12.







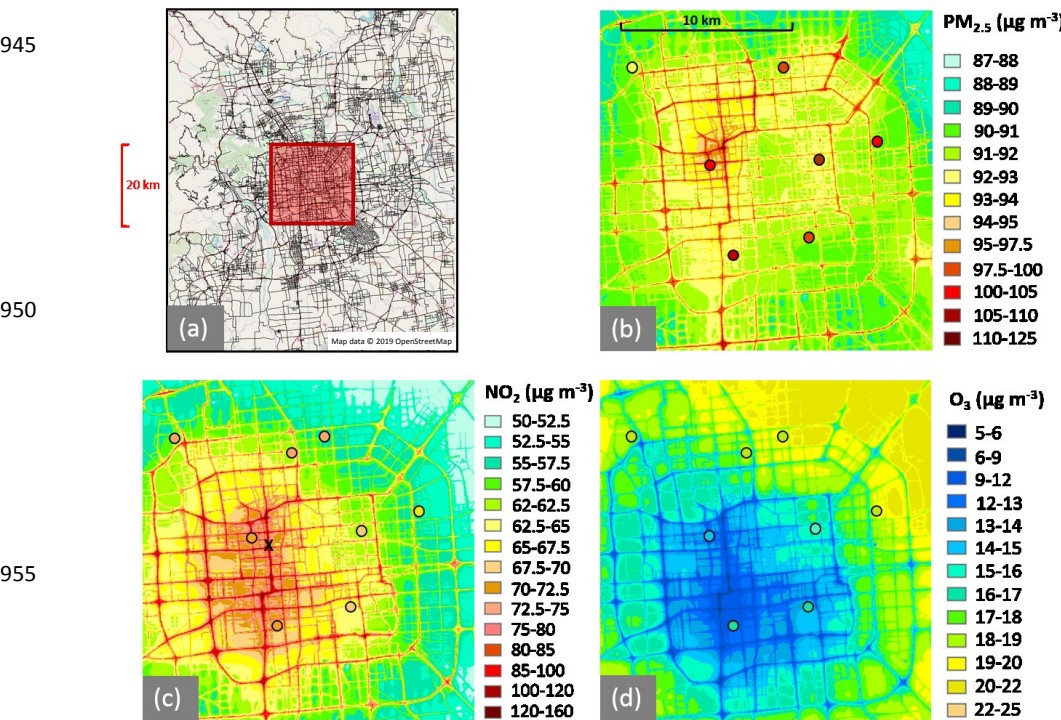

**Figure 4: Spatial maps of mean PM₂.₅ (b), NO₂ (c), and O₃ (d) concentrations for the winter campaign period (5/11/16 to 10/12/16), simulated using the MEIC Opt emissions inventory. Simulated concentrations cover the region marked in (a) (© OpenStreetMap contributors 2019. Distributed under a Creative Commons BY-SA License). Mean measured concentrations at monitoring network sites (NO₂, O₃ and PM₂.₅) and the IAP field site (NO₂ and O₃) are represented by coloured dots.**

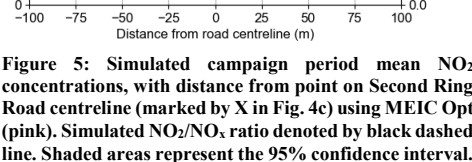

**Figure 5: Simulated campaign period mean NO₂ concentrations, with distance from point on Second Ring Road centreline (marked by X in Fig. 4c) using MEIC Opt (pink). Simulated NO₂/NOₓ ratio denoted by black dashed line. Shaded areas represent the 95% confidence interval.**





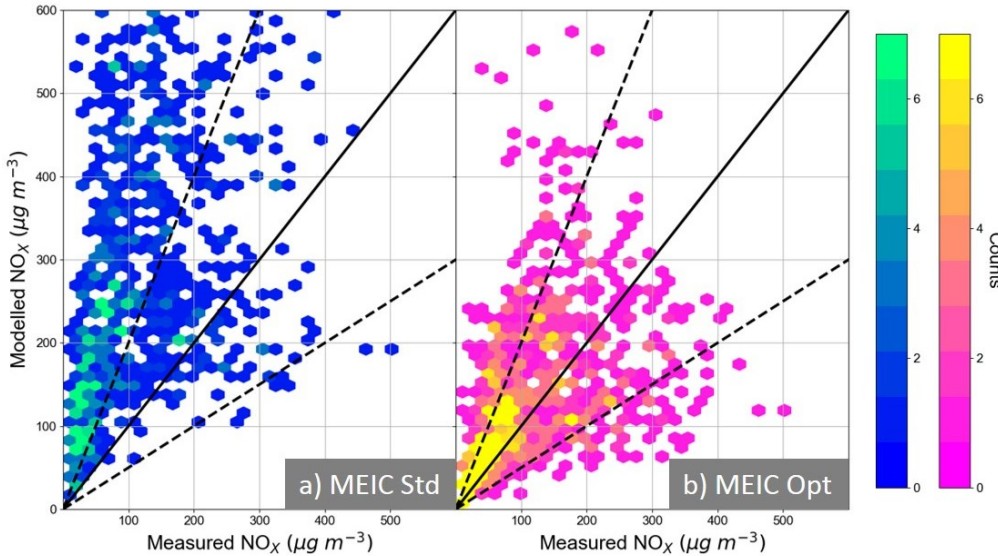

**Figure 6: Hourly measured and modelled NO$_x$ concentrations during the campaign period at the IAP field site. Panel (a) and (b) showing concentrations simulated using MEIC Std and MEIC Opt, respectively. Colours represent the total number of matching hourly measured and modelled values contained within distinct hexagonal bins. Dashed lines mark factor of two difference between measured and simulated concentrations.**






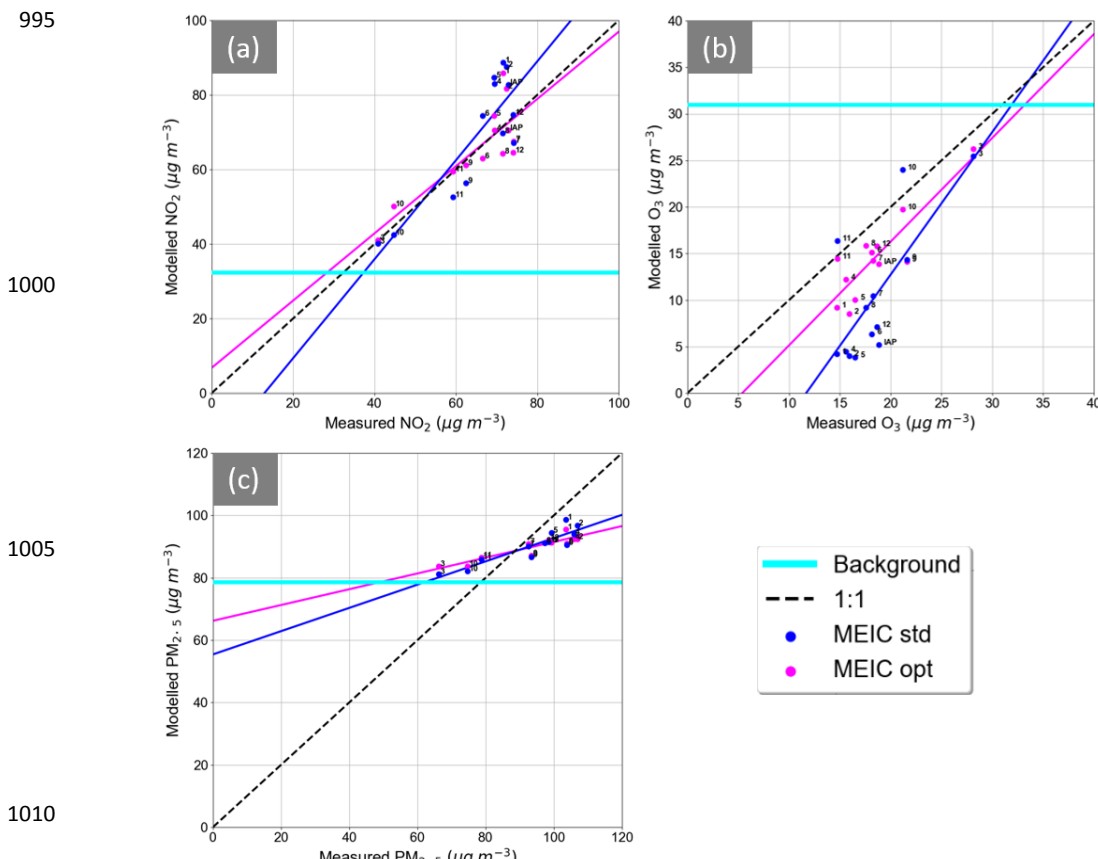

**Figure 7: Campaign period mean measured and modelled (a) NO₂, (b) O₃, and (c) PM₂.₅ concentrations at all monitoring network sites (numbered) and the IAP field site (NO₂ and O₃). Blue and pink lines indicate concentrations simulated using MEIC Std and MEIC Opt, respectively. Horizontal light blue line represents campaign period mean background concentrations calculated from measurements.**






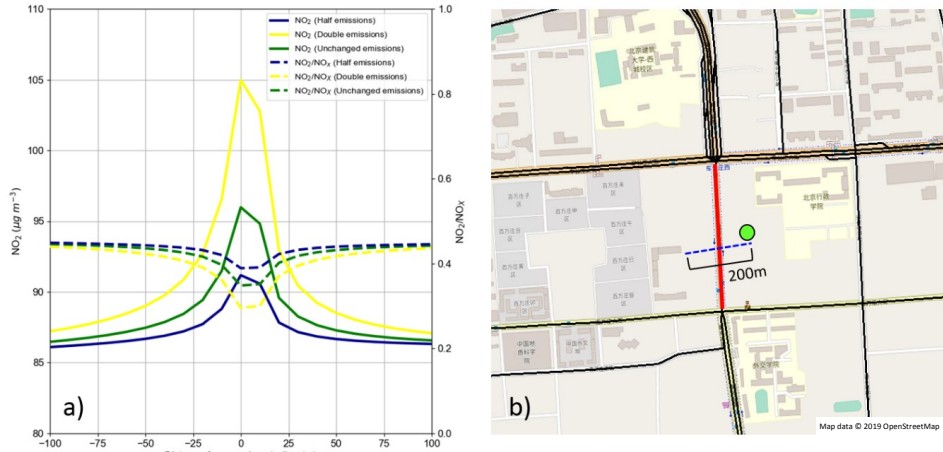

**Figure 8: (a) Campaign period mean simulated NO₂ concentrations and NO₂/NOₓ concentration ratios with distance from road centre along cross-road slice marked in (b) using half (blue), double (yellow) and unchanged (green) emissions of all species from explicit road source marked by red line. Green circle in (b) marks position of monitoring site 1. (b) © OpenStreetMap contributors 2019. Distributed under a Creative Commons BY-SA License.**

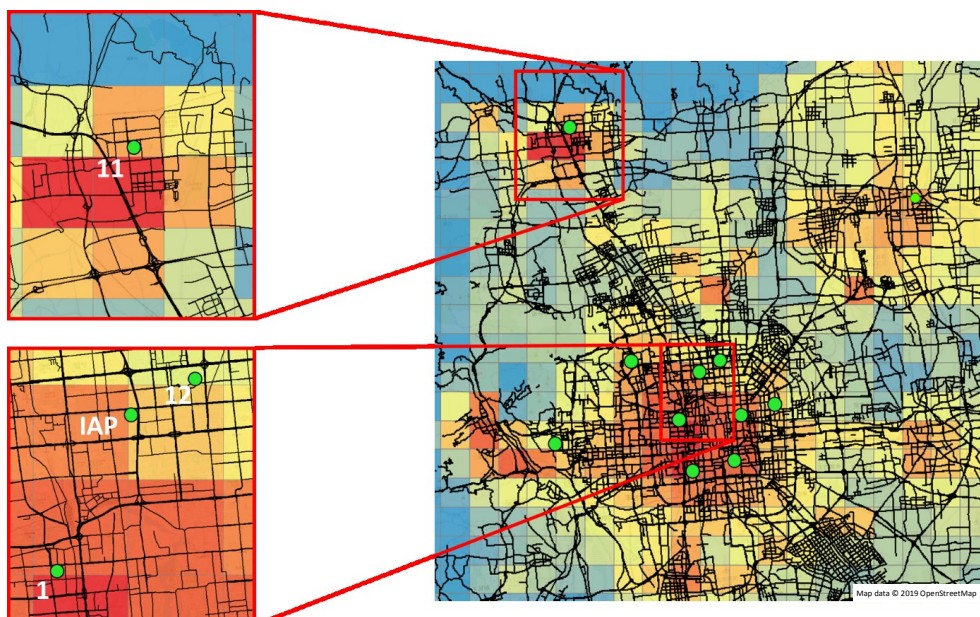

**Figure 9: Spatial distribution of November and December mean MEIC Opt NO₂ emissions (all emission sectors) overlaid with Beijing road network (source: OpenStreetMap). Enlarged regions cover urban sites 1, 12 and IAP as well as suburban site 11. Right panel: © OpenStreetMap contributors 2019. Distributed under a Creative Commons BY-SA License.**







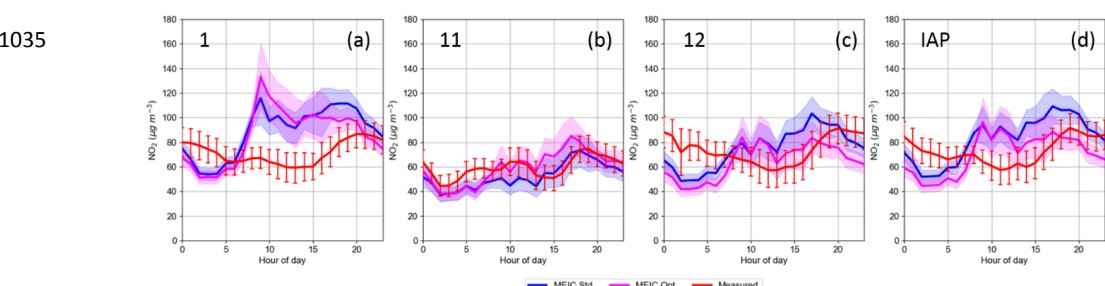


**Figure 10: Campaign period mean diurnal variation in modelled and measured NO₂ concentrations at sites (a) 1, (b) 11, (c) 12, and (d) IAP. Modelled concentrations produced using both MEIC Std (blue) and MEIC Opt (pink). Measurements marked by red line. Shaded areas and error bars represent the 95% confidence intervals for simulated and measured concentrations, respectively.**


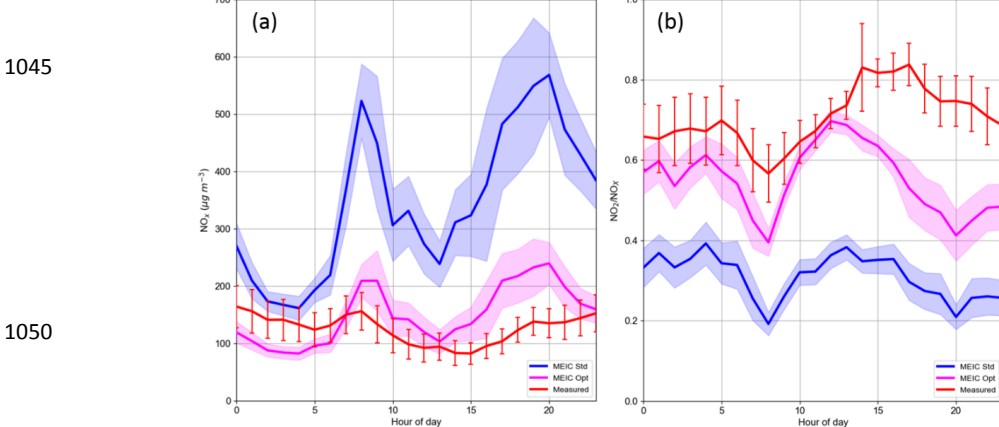


**Figure 11: Campaign period mean diurnal variation in modelled and measured (a) NOₓ concentrations and (b) NO₂/NOₓ concentration ratios at the IAP field site. Modelled concentrations produced using both MEIC Std (blue) and MEIC Opt (pink). Measurements marked by red line. Shaded areas and error bars represent the 95% confidence intervals for simulated and measured concentrations, respectively.**







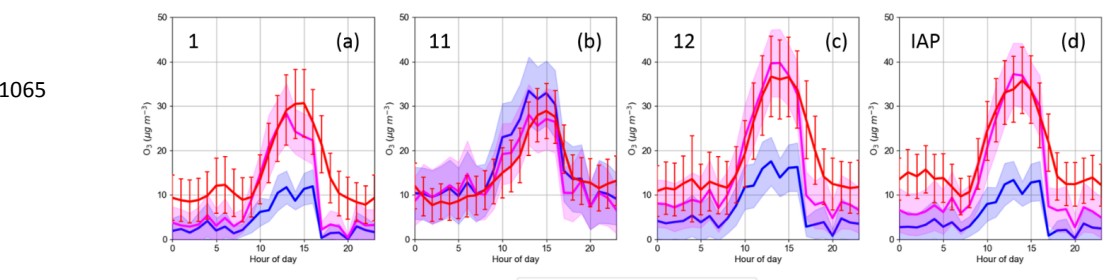

**Figure 12: Campaign period mean diurnal variation in modelled and measured O₃ concentrations at sites (a) 1, (b) 11, (c) 12 and (d) IAP. Modelled concentrations produced using both MEIC Std (blue) and MEIC Opt (pink). Measurements marked by red line. Shaded areas and error bars represent the 95% confidence intervals for simulated and measured concentrations, respectively.**

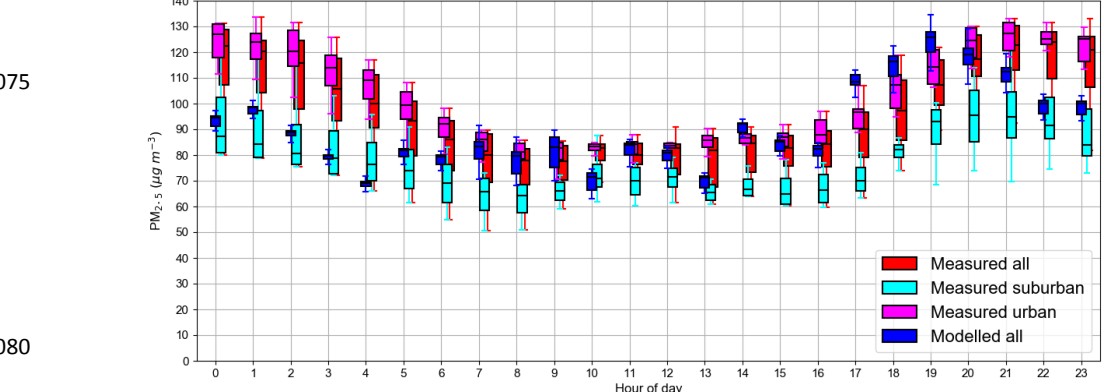

**Figure 13: Variations in site-specific campaign period mean measured (red) and modelled (blue) PM₂.₅ concentrations across all monitoring network stations for each hour of the day. Measurements sub-divided to highlight the variation between suburban (cyan) and urban (pink) monitoring network stations specifically.**



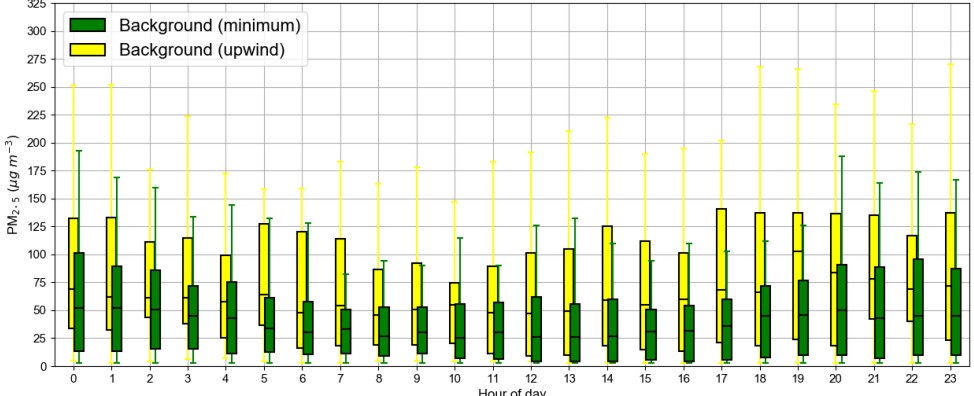

**Figure 14: Ranges in campaign period mean calculated background PM$_{2.5}$ concentrations for each hour of the day using minimum (green) and upwind (yellow) concentration methodologies.**

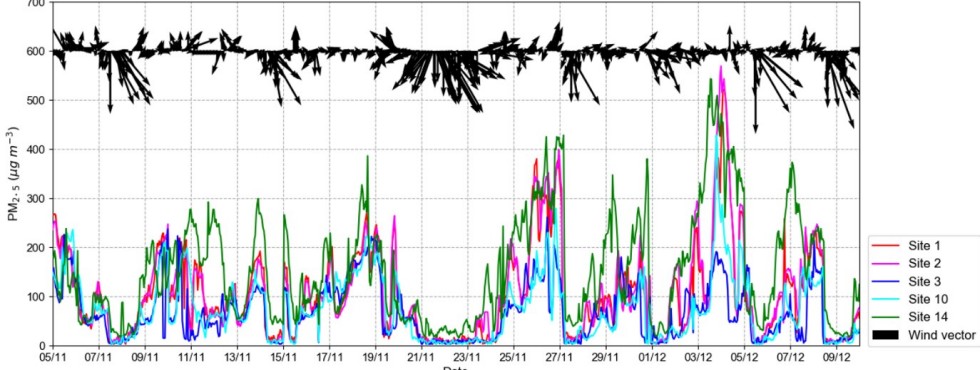

**Figure 15: Hourly PM$_{2.5}$ concentrations at measurement sites 1, 2, 3, 10 and 14 during the campaign period. Wind vectors, representing wind speed magnitude and direction recorded at the airport meteorological station, are also provided (black arrows).**





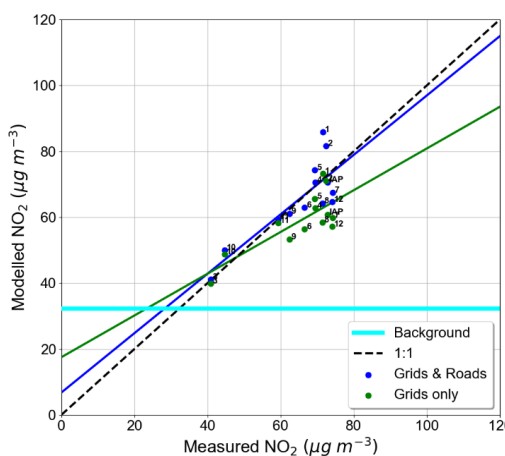

**Figure 16: Campaign period measured and modelled NO₂ concentrations at all measurement sites (numbered). Modelled concentrations produced using 3-D grid and explicit road emission sources (blue), and 3-D grid sources only (green) derived from the MEIC Opt emissions inventory. Horizontal light blue line represents campaign period mean background NO₂ concentrations calculated from measurements.**

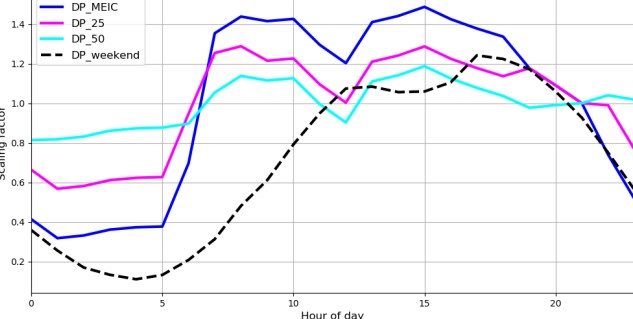

**Figure 17: Diurnal emissions profiles applied to the simulations shown in Fig. 18. Standard MEIC diurnal emissions profile (DP_MEIC) marked by blue line; modified DP_MEIC with increased proportions of nighttime emissions marked by pink (DP_25) and cyan (DP_50) lines and weekend emissions profile marked by dashed black line.**



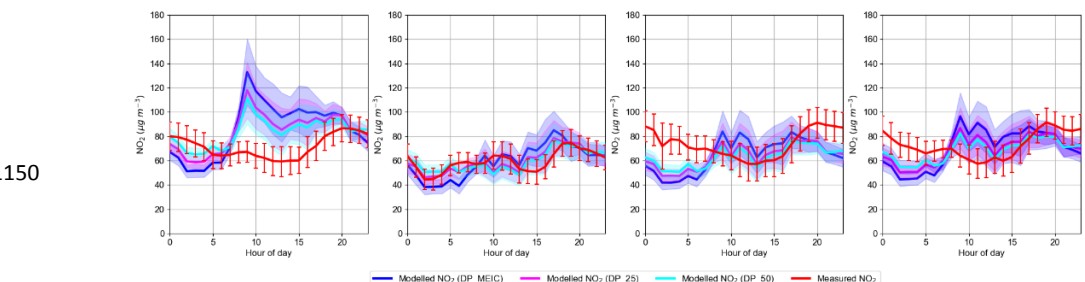

**Figure 18: Campaign period mean diurnal variation in measured and modelled NO₂ concentrations using MEIC Opt at sites (a) 1, (b) 11, (c) 12, and (d) IAP. Measurements marked by red line. Shaded areas and error bars represent the 95% confidence intervals for simulated and measured concentrations, respectively.**

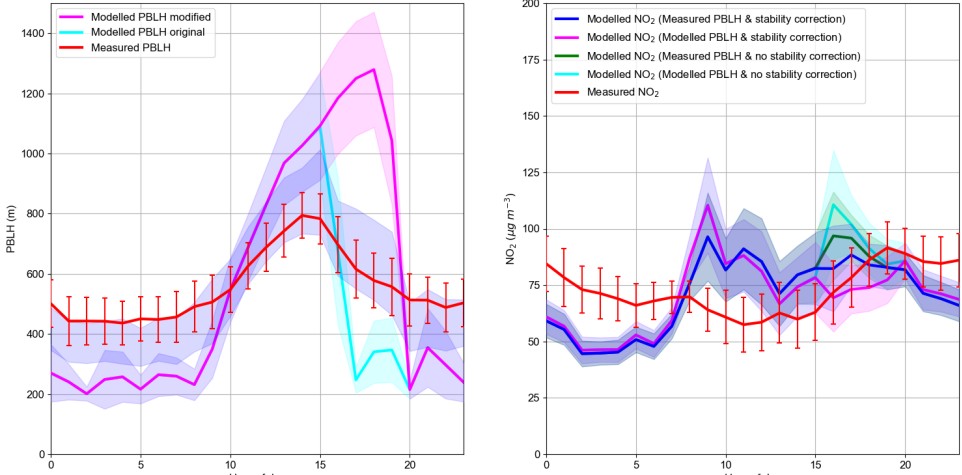

**Figure 19: (a) Campaign period mean diurnal variation in modelled PBLH with stability correction (pink), modelled PBLH without stability correction (cyan) and measured PBLH (red). (b) Campaign period mean diurnal variation in measured (red) and modelled NO₂ concentrations with measured PBLH and stability correction (blue), modelled PBLH with stability correction (pink), measured PBLH without stability correction (green), and modelled PBLH without stability correction (cyan) at the IAP field site. Shaded areas and error bars represent the 95% confidence intervals for simulated and measured PBLH and concentrations, respectively.**


| | Site name | Site type | Latitude (°N) | Longitude (°E) | Distance to nearest road centreline (m) | Nearest Road type |
|---|---|---|---|---|---|---|
| 1 | Guanyuan | Urban | 39.93 | 116.34 | 90 | Secondary |
| 2 | Wanshou Xigong | Urban | 39.88 | 116.35 | 80 | Tertiary |
| 3 | Dingling | Suburban | 40.29 | 116.22 | 285 | Tertiary |
| 4 | Dongsi | Urban | 39.93 | 116.42 | 200 | Secondary |
| 5 | Tiantan | Urban | 39.89 | 116.41 | 90 | Tertiary |
| 6 | Nongzhanguan | Urban | 39.94 | 116.46 | 400 | Trunk |
| 7 | Haidan Wanliu | Urban | 39.99 | 116.29 | 100 | Tertiary |
| 8 | Gucheng | Urban | 39.91 | 116.18 | 260 | Tertiary |
| 9 | Shunyicheng | Suburban | 40.13 | 116.66 | 190 | Secondary |
| 10 | Huairouzhen | Suburban | 40.33 | 116.63 | N/A | Tertiary |
| 11 | Changpingzhen | Suburban | 40.22 | 116.23 | 200 | Secondary |
| 12 | Aoti Zhongxin (Olympic Park) | Urban | 39.98 | 116.4 | 110 | Secondary |
| 13 | IAP | Urban | 39.97 | 116.37 | 110 | Secondary |
| 14 | TCM Medical Material Company | Urban | 39.52 | 116.69 | N/A | Secondary |

Table 1: Locations (latitude and longitude) of all monitoring stations, including distinction between urban (within sixth ring road) and suburban site types. Approximate distance (nearest 10 m) from each monitoring station to nearest road centreline and corresponding road type also provided.

| Campaign period mean aggregate pollutant emission rates (Tonnes day$^{-1}$) | | | | | | | | | | | | |
|---|---|---|---|---|---|---|---|---|---|---|---|---|
| Region of domain | NO$_2$ | | NO$_x$ | | PM$_{2.5}$ | | PM$_{10}$ | | SO$_2$ | | VOC | |
| | S | O | S | O | S | O | S | O | S | O | S | O |
| All | 60.2 | 46.9 | 889.3 | 504.6 | 86.2 | 110.3 | 156.5 | 176.1 | 72.5 | 54.4 | 717.6 | 1942.5 |
| Change (%) | -22.1 | | -43.3 | | 28.0 | | 12.5 | | -25.0 | | 170.7 | |
| Urban | 44.0 | 22.1 | 649.1 | 238.3 | 49.9 | 46.9 | 89.3 | 69.5 | 42.4 | 27.5 | 476.8 | 1273.1 |
| Change (%) | -49.8 | | -63.3 | | -6.0 | | -22.2 | | -35.1 | | 167.0 | |
| Suburban | 16.3 | 24.7 | 240.2 | 266.2 | 36.3 | 63.4 | 67.2 | 106.7 | 30.1 | 27.0 | 240.8 | 669.4 |
| Change (%) | 51.5 | | 10.8 | | 74.7 | | 58.8 | | -10.3 | | 178.0 | |

Table 2: Campaign period mean MEIC Std (S) and MEIC Opt (O) pollutant species emissions (Tonnes day$^{-1}$) aggregated across all, urban and suburban grid cells. Change (%) in emissions between inventories, following optimisation, calculated as (O − S/S) x 100.



| Road Type | Weighting |
|-----------|-----------|
| Motorway | 0.7 |
| Trunk | 0.5 |
| Primary | 0.4 |
| Secondary | 0.25 |
| Tertiary | 0.15 |

**Table 3: Estimated emission weighting factors for each modelled road type.**

| | | Mean Concentrations (µg m$^{-3}$) | | | Model Evaluation Statistics | | | | | |
|---|---|---|---|---|---|---|---|---|---|---|
| | | | | | NMSE | | Fb | | R | |
| | Sites | Mod (S) | Mod (O) | Obs | S | O | S | O | S | O |
| PM$_{2.5}$ | All | 90.3 | 89.8 | 93.4 | 0.37 | 0.37 | -0.03 | -0.04 | 0.76 | 0.76 |
| | Urb | 93.4 | 92.1 | 100.9 | 0.36 | 0.36 | -0.08 | -0.09 | 0.78 | 0.78 |
| | Sub | 84.0 | 85.3 | 78.3 | 0.40 | 0.41 | 0.07 | 0.09 | 0.74 | 0.74 |
| O$_3$ | All | 10.4 | 14.6 | 18.5 | 1.54 | 0.74 | -0.56 | -0.24 | 0.71 | 0.79 |
| | Urb | 6.1 | 12.8 | 17.2 | 3.20 | 0.93 | -0.95 | -0.29 | 0.70 | 0.77 |
| | Sub | 20.0 | 18.6 | 21.4 | 0.48 | 0.47 | -0.07 | -0.14 | 0.82 | 0.83 |
| NO$_2$ | All | 69.5 | 65.7 | 65.3 | 0.27 | 0.30 | 0.06 | 0.00 | 0.55 | 0.53 |
| | Urb | 79.2 | 71.4 | 71.3 | 0.27 | 0.31 | 0.10 | 0.00 | 0.42 | 0.44 |
| | Sub | 47.9 | 52.9 | 51.8 | 0.21 | 0.23 | -0.08 | 0.02 | 0.74 | 0.70 |
| NO$_x$ | IAP | 345.5 | 149.8 | 126.1 | 2.35 | 0.63 | 0.93 | 0.17 | 0.35 | 0.41 |

**Table 4: Statistical evaluation of modelled pollutant concentrations for the campaign period, using MEIC Std (S) and MEIC Opt (O) emissions inventories. Mean modelled (Mod) and observed (Obs) concentrations and statistics divided into all (12 monitoring network sites and IAP field site for NO$_2$ and O$_3$, monitoring network sites only for PM$_{2.5}$) and urban and suburban monitoring site groups. Urban and suburban sites defined in Table 1. NO$_x$ measurements only available at the IAP field site. Mean concentrations and statistics calculated from matching hourly values.**

| | | Mean Concentrations (µg m$^{-3}$) | | | Model Evaluation Statistics | | | | | |
|---|---|---|---|---|---|---|---|---|---|---|
| | | | | | NMSE | | Fb | | R | |
| | Site | Mod (G) | Mod (G-R) | Obs | G | G-R | G | G-R | G | G-R |
| NO$_2$ | All | 58.9 | 65.7 | 65.3 | 0.28 | 0.30 | -0.10 | 0.00 | 0.59 | 0.53 |
| | Urb | 62.8 | 71.4 | 71.3 | 0.29 | 0.31 | -0.13 | 0.00 | 0.51 | 0.44 |
| | Sub | 50.1 | 52.9 | 51.8 | 0.21 | 0.23 | -0.03 | 0.02 | 0.73 | 0.70 |

**Table 5: Same information presented as in Table 4 but for NO$_2$ concentrations simulated (MEIC Opt) using 3-D grid sources only (G) as well as 3-D grid and explicit road sources (G-R) (also presented in Table 4).**
