# Peer review of "Street-scale air quality modelling for Beijing during a winter 2016 measurement campaign"

_Atmospheric Chemistry and Physics, 2019_

## Referee Comment (RC1) · Anonymous Referee #1 · 24 Oct 2019

In this manuscript, the authors use the Gaussian pollution dispersion and a chemistry model named ADMS-Urban measured the street-scale resolution concentrations of NOx, NO2, O3 and PM2.5 in Beijing. They construct a traffic emissions inventory, and this method improves the consistency of simulation data and measurement data of Beijing's air quality monitoring network and the Institute of Atmospheric Physics (IAP) field site. ADMS-Urban model can solve the sharp concentration gradients adjacent to major roads. This manuscript can provide valuable information for evaluating the street-scale air quality. The following advices hope to attract your attention.

Q1: Several articles have been quoted many times in the introduction. It is recommended that multiple references be cited to reflect the amount of reading and to enhance the persuasion.

[Figure]

Q2: The novelty of this work should be highlighted in the introduction. I suggest present in the last two paragraph.

Q3: The introduction is long and unclear. You should summarize the advantages and disadvantages of various methods at the end to highlight your innovation points of the article.

Q4: Line 134, don't quote the two references separately, insert them at the end of the sentence.

Q5: Line 144, please explain the reason for using hourly wind direction and speed.

Q6: Lines 229-231, datas should be provided to support the point.

Q7: It is suggested that the idea of Sect. 3.3 should focuses on the comparison of the simulation results of the MEIC-Std and MEIC-Opt two emissions inventories. On this basis, the specific conclusions are explained.

Q8: The title should be placed on the top of the table, as shown in Table 1 in the 1163 line. Please correct it in sequence.

Q9: How to modify the PBL stability parameters should be detailed in Sect. 2.1.2 rather than just in results , and should referenced with the conclusion.

Q10: Note some details to the format of the paper. Such as: Please change the font in formula 4ïïjĹLine 305ïïjĽinto italics. The last name in the legend in Fig 1. is incomplete. The ":" and "." behind Fig 1. and Fig 2. should be consistent.

Q11: Line 560 "The corresponding Fb value improvement, at urban sites, from. . . . . .." It is should tell the reader more clear that why the Fb can reflect the traffic emission (for different range)? And some references should be cited.

Q12: Line 570 "Little change is seen across suburban areas with the inclusion of explicit road source emissions, reflecting the lower density of roads and more dominant con-tribution from diffuse emissions with distance away from Beijing's urban centr" Please

make more explaination.

Q13: Line 620: "The results suggest that although atmospheric stability has a strong impact on NO2 concentrations, the use of observed PBLH instead of modelled heights has little effect". This conclusion should be extended.

---

## Referee Comment (RC2) · Anonymous Referee #2 · 4 Nov 2019

In this paper, Biggart et al presented a street-level air quality simulation study for Beijing urban area using an urban air pollution dispersion and chemistry model ADMS-Urban. The predictions were evaluated against observations during winter campaign in 2016 in the same area. Several sensitivity tests were conducted to investigate possible reasons for the discrepancies between model and observations. Studies like this provide useful information on high-resolution air quality simulation in complex urban areas. The paper is generally well written. A few commons are provided below.

1. The model domain for ADMS-Urban is 75 km x 90 km. Author mentioned that the model has street-level resolution, what is exact resolution setting in the model in terms of meters? Does the resolution vary between different land uses, eg. Road and other areas? If they all use the same resolution as road, then it would require

[Figure]

a very high computational demand. 2. Page 6 Line 225, Measured concentrations at 12 of the 35 monitoring stations in Beijing were used to produce the background concentration. Is that a specific reason to exclude the other 23 stations? 3. Page 7 Line 245, agricultural emission is not included in the model simulation? 4. Page 7 Line 250, emissions for the industrial and residential sectors were distributed based on Gross Domestic Product (GDP) and population density to grid-level resolutions? What's the resolution for this? Is information of GDP and population density available at your grid-level resolutions? 5. Table 3 provides the weighting factors for road emissions. Are the weighting factors the same for all pollutants that simulated in this study? 6. Page 8 Line 290, the urban areas are more congested than suburban. This study used the same weighting factors for urban and suburban, which may underestimate the urban emissions. It is recommended to try different weighting factors for urban and suburban and test the impacts on simulation results. 7. Figure 16: suggest to change the color scheme for "Grid & Roads" and "Grid only". It is very hard to distinguish them on the map.

───────────────────────────

---

## Author Comment (AC1) · 24 Jan 2020

We thank both reviewers for their detailed and insightful feedback on this study which has considerably improved the manuscript. Responses to each comment are structured as follows: (a) reviewer comment (in bold), (b) our response to the comment, (c) changes to the manuscript (in quotation marks and italics). In the revised manuscript modified text is highlighted using Track Changes.

Referee #1

In this manuscript, the authors use the Gaussian pollution dispersion and chemistry model named ADMS-Urban simulating the street-scale resolution concentrations of NOx, NO2, O3 and PM2.5 in Beijing. They construct a traffic emissions inventory, and

this method improves the consistency of simulation data and measurement data of Beijing's air quality monitoring network and the Institute of Atmospheric Physics (IAP) field site. ADMS-Urban model can solve the sharp concentration gradients adjacent to major roads. This manuscript can provide valuable information for evaluating the street-scale air quality. The following advices hope to attract your attention.

We thank the reviewer for their positive feedback.

1. Several articles have been quoted many times in the introduction. It is recommended that multiple references be cited to reflect the amount of reading and to enhance the persuasion.

We thank the reviewer for this comment as we agree some articles have been cited on a number of occasions. To address this, the following references have been added throughout the introduction to help explain key points:

On Page 2 Line 55 studies by Li et al. (2018) and Cui et al. (2019) are cited in which data from Beijing's air quality monitoring network is extensively analysed. Cheng et al. (2019) has been removed as it is more relevant to emission reduction estimates in Beijing discussed in the following paragraph, in which it is referenced multiple times. In the subsequent paragraph on Line 63 a reference to Sun et al. (2018), who investigated China's emission trends, has been added. On Line 65 articles by Ni et al. (2018) and Wang et al. (2019) are included in which the impacts of China's extensive implementation of emission control techniques are discussed.

As described in detail in response to reviewer comment 3, on Page 3 Line 86 and 89 recent studies describing alternative urban air quality modelling techniques are added – J. Xu et al. (2019), M. Xu et al. (2019) and Lugon et al. (2019). Page 2/3 Line 84-89 now reads:

"Land use regression (LUR) modelling studies, combining geospatial indicators with air quality measurement data, can generate local scale (<1 km) pollutant level variations,

but have been limited by the sparsity of monitoring network data available in Beijing (J. Xu et al. 2019; M. Xu et al. 2019). Alternatively, box models, such as The Model of Urban Network of Intersecting Canyons and Highways (MUNICH), are used to calculate pollutant concentrations within street canyons, but require detailed information on the spatial dimensions of a city's street canyons and are restricted by assumptions of uniform concentrations along individual road segments (Lugon et al. 2019)."

On Line 106 a further study (Zhang et al. 2018) describing a recently constructed bottom-up road traffic emissions inventory for Beijing is cited. Page 3 Line 106-108 of the updated manuscript:

" A bottom-up street-scale vehicle emissions inventory was also created by Zhang et al. (2018), using traffic surveys and video identification of vehicle fleet composition to evaluate the impact of a new low emission zone (LEZ) in urban Beijing."

2. The novelty of this work should be highlighted in the introduction. I suggest present in the last two paragraph.

We thank the reviewer for this comment. We have now highlighted the novelty of this work in relation to both the methodology for generating the road emissions network, which can be applied to urban areas elsewhere with limited data availability, and the evaluation of street-scale modelled concentrations using measurements from both Beijing's air quality monitoring network and an intensive measurement campaign. Page 3 Line 111-113 of the updated manuscript:

"However, this work provides a robust framework suitable for similar street-scale air quality modelling across large urban areas with limited data availability that future human health studies can build on."

Page 3 Line 118-120 of the updated manuscript:

"Measured pollutant concentrations from both the APHH-China campaign and Beijing's air quality monitoring network are used to evaluate modelled concentrations, providing

valuable insight into the key processes that impact street-scale air quality."

The suitability of ADMS-Urban for performing sensitivity studies that further explore the discrepancies between modelled and measured concentrations is also highlighted on Page 3 Line 120:

"The adaptability of ADMS-urban is utilised in a series of further sensitivity simulations aimed at exploring the impact that..."

3. The introduction is long and unclear. You should summarize the advantages and disadvantages of various methods at the end to highlight your innovation points of the article.

We thank the reviewer for this comment. We have now summarised the advantages and disadvantages of various methods by outlining both land use regression (LUR) and box model approaches to urban air quality modelling, with limitations of both highlighted.

The manuscript has been updated on Page 2 from Line 83:

"...As a result, a range of street-scale resolution air quality modelling techniques have recently emerged. Land use regression (LUR) modelling studies, combining geospatial indicators with air quality measurement data, can generate local scale (<1 km) pollutant level variations, but have been limited by the sparsity of monitoring network data available in Beijing (J. Xu et al. 2019; M. Xu et al. 2019). Alternatively, box models, such as The Model of Urban Network of Intersecting Canyons and Highways (MUNICH), are used to calculate pollutant concentrations within street canyons, but require detailed information on the spatial dimensions of a city's street canyons and are restricted by assumptions of uniform concentrations along individual road segments (Lugon et al. 2019)."

Distinction between ADMS-Urban and the US EPA's environmental regulatory model AERMOD has been added with ADMS-Urban's use of a simplified chemistry scheme

explicitly stated on Page 3 Line 93:

"The additional modelling of local fast chemistry processes on pollutant emissions with ADMS-Urban, involving the simplified Generic Reaction Set (GRS) chemistry scheme, including NOx-O3 reactions, enables sharp concentration gradients adjacent to major urban sources to be captured (Hood et al. 2018)."

As outlined above in response to reviewer 1 comment 1, on Line 107 an additional study by Zhang et al. (2018) is cited in which the impacts of a recently introduced low emission zone (LEZ) in urban Beijing on a newly constructed bottom-up street-scale road traffic emissions inventory are investigated. On Line 106-108 of the updated manuscript:

"A bottom-up street-scale vehicle emissions inventory was also created by Zhang et al. (2018), using traffic surveys and video identification of vehicle fleet composition to evaluate the impact of a new low emission zone (LEZ) in urban Beijing."

The disadvantages of the methodology adopted here for generating the explicit network of road emissions compared with the Zhang et al. (2018) and Yang et al. (2019) studies have been outlined on Page 3 Line 110:

"Unlike the data-intensive methodology adopted by Yang et al. (2019), spatiotemporal variations in traffic volume and vehicle type are not considered here."

However, the advantage of apportioning coarser resolution gridded emissions onto the openly available OpenStreetmap (OSM) spatial road network compared with the data-intensive Zhang et al. (2018) and Yang et al. (2019) techniques is highlighted on Page 3 Line 111:

"...However, this work provides a robust framework suitable for similar street-scale air quality modelling across large urban areas with limited data availability that future human health studies can build on."

The reviewer also commented on the introduction's excessive length. To address this,

the previously detailed summaries of numerous regional modelling studies in Beijing have been removed. Page 2 Lines 80-82 of the updated manuscript:

"Numerous regional modelling studies, incorporating emission inventories such as MEIC and Eulerian chemical transport models (CTMs), have been carried out for Beijing (Liu et al. 2016; Petaja et al. 2016; Li et al. 2017; Wang et al. 2017; Wang et al. 2018; Chang et al. 2019)."

Descriptions of earlier urban air quality modelling studies using ADMS-Urban in other countries have also been removed from Page 3 in order to allow for greater focus on the advantages and disadvantages of distinctly different approaches to urban air quality modelling and road traffic emission inventory construction.

4. Line 134, don't quote the two references separately, insert them at the end of the sentence.

As suggested, both references have been moved to the end of the sentence.

Page 4 Line 135-138 of the manuscript:

"ADMS-Urban, developed by Cambridge Environmental Research Consultants (CERC), is a quasi-Gaussian pollution dispersion and chemistry model that has been applied worldwide for environmental regulation, investigation and assessment of emission control strategies and generation of high spatial resolution air quality forecasts (McHugh et al. 2005; Carruthers, 2009; Cai and Xie, 2011)."

5. Line 144, please explain the reason for using hourly wind direction and speed.

Hourly values of all meteorological variables (wind speed, wind direction, air temperature and cloud cover ) are used to drive hourly plume dispersion calculations. Parameters determining the stability of the PBL ($F\theta 0$, $U^*$ and LMO) are calculated for each hour using the input meteorology. These parameters are subsequently used in the model to calculate horizontal and vertical plume spread parameters that determine hourly pollutant concentrations via Gaussian distribution equations and the summation

of contributions from individual emission sources.

To clarify that all input meteorological variables used are hourly-varying, Page 4 Line 146 of the manuscript is updated to:

"For this study, we use hourly wind speed, wind direction, air temperature and cloud cover data from the Beijing Capital International Airport Meteorology Observatory..."

Line 160-162 of the updated manuscript:

"This PBLH/LMO parameterisation controls the vertical and horizontal spread extents of each emitted Gaussian plume, with the aggregate contribution from each individual emission source determining hourly simulated pollutant concentrations."

6. Lines 229-231, data should be provided to support this point.

The manuscript has been updated to explicitly state the monitoring stations used to estimate background PM2.5 and PM10 concentrations as well as the wind directions associated with each. Page 6 Line 234-236:

"For particulate matter (PM2.5 and PM10), an hourly upwind background concentration is derived based on wind direction with concentrations selected from sites 3 (270-360o), 10 (0-90o) and 14 (90-270o) located to the NW, NE and SE of urban Beijing, respectively."

7. It is suggested that Sect. 3.3 should focus on the comparison of the simulation results using the MEIC Std and MEIC Opt emission inventories. On this basis, the specific conclusions are explained.

We thank the reviewer for this comment, however the exact nature of the recommended changes is unclear. Both Sect. 3.2 and 3.3 present comparisons of simulation results using the MEIC Std and MEIC Opt inventories, however with different objectives. Through the statistical evaluation of site-specific period mean measured and simulated concentrations, the primary aims of Sect. 3.2 are to demonstrate overall model

performance as well as the impact of the redistribution and magnitude adjustment of emissions with MEIC Opt on the spatial variation of simulated pollutant concentrations. Clear improvements in model performance using MEIC Opt for all pollutant species highlight issues with using spatial proxy-based emission inventories for street-scale air quality modelling. In Sect. 3.3 the focus is instead on learning how well the model is capturing the interaction between diurnally varying emissions and PBL dynamics, driven by meteorology and surface characteristics. This provides information on possible missing sources in the emission inventories, dominant chemical processes and the impact of the urban heat island (UHI).

We feel that Sect. 3.2 and 3.3 offer different and valuable information on the results using the MEIC Std and MEIC Opt inventories that future urban air quality modelling studies can build on and therefore should retain their current structure.

8. The title should be placed on the top of the table, as shown in Table 1 on Line 1163. Please correct it in sequence.

As suggested by the reviewer, each table caption has been placed above the table.

9. How to modify the PBL stability parameters should be detailed in Sect. 2.1.2 rather than just in results, and should be referenced with the conclusion.

We agree and have added further details describing the methodology for adjusting the PBL stability parameters. On Page 5 Line 191-195 of the updated manuscript:

"To account for this, a constant rate of decrease of PBLH/LMO has been assumed between original modelled values for 3 pm and 8 pm, producing the modified campaign period mean PBLH/LMO diurnal profile illustrated in Fig. 2. Modified LMO values from 4-7 pm are added to the set of input meteorological variables for all subsequent simulations, with the directly input PBLH measurements remaining unchanged."

A reference to this description has been added to Page 16 Line 637: "The early evening stability adjustment (Sect. 2.1.2)..."

[Figure]

10. Note some details to the format of the paper. Such as: Please change the font in formula 4ïïj′LLine 305ïïjĽinto italics. The last name in the legend in Fig 1. is incomplete. The ":" and "." behind Fig 1. and Fig 2. should be consistent.

For Fig 1. we have not altered the "Beijing Capital International Airport" label but have instead removed the "meteorology observatory" label in the caption. We have made the other formatting edits as suggested by the reviewer.

11. Line 560 "The corresponding Fb value improvement, at urban sites, from....." It is should tell the reader more clearly how the Fb can reflect the traffic emission (for different range)? And some references should be cited.

We thank the reviewer for this comment and have further explained how Fb improvement is related to the inclusion of explicitly represented traffic sources through reference to the concentration gradients presented in Fig. 4. Two other ADMS-Urban studies are cited (Dédélé and Miskinyté, 2015; Hood et al. 2018) in which Fb improvements are associated with enhanced traffic emissions from explicit road sources.

Page 15 Line 579-584 of the updated manuscript now reads:

"This modelled urban $NO_2$ concentration increase results in a Fb value improvement from -0.13 to 0 (Table 5) reflecting the greater $NO_2$ levels simulated by the model at locations in close proximity to explicit roads. By using grid sources only, the road traffic emissions are diluted over each 3 x 3 km grid cell and the strong concentration gradients associated with a region densely populated by major roads, illustrated in Fig. 4, are not captured. Similarly, Dédélé and Miskinyté (2015) and Hood et al. (2018) found that increased traffic emissions due to higher traffic volume and adjusted emission factors, respectively, produced improved Fb values using ADMS-Urban."

12. Line 570: "Little change is seen across suburban areas with the inclusion of explicit road source emissions, reflecting the lower density of roads and more dominant contribution from diffuse emissions with distance away from Beijing's urban centre". Please

make more explanation.

We have included additional references to model evaluation statistics (Line 593) in order to emphasise the lack of influence of explicit road sources on suburban NO2 concentrations compared with urban locations. The strong concentration gradients influencing near-road urban locations are not present across suburban areas, with diffuse emissions from residential sources having a stronger influence. Two studies investigating the importance of residential heating and cooking emissions from coal combustion in Beijing are now cited (Cai et al. 2018; Li et al. 2018).

Page 15 Line 593-599 of the updated manuscript now reads:

"Minimal R value changes and a much lower Fb improvement, from -0.03 to 0.02, are seen across suburban compared to urban areas, following the inclusion of explicit road source emissions. This reflects the lower density of roads in suburban areas (Fig. 4) and therefore absence of the strong concentration gradients that enhance NO2 levels at near-road urban locations. The relative influence of diffuse emissions contained within the underlying gridded emission sources on simulated pollutant concentrations is therefore more prominent with distance from Beijing's urban centre, with previous studies specifically highlighting the persisting importance of residential coal combustion for heating and cooking during winter in suburban and rural Beijing (Cai et al. 2018, Li et al. 2018)."

13. Line 620: "The results suggest that although atmospheric stability has a strong impact on NO2 concentrations, the use of observed PBLH instead of modelled heights has little effect". This conclusion should be extended.

This conclusion has been extended to more clearly highlight the small impact of PBLH changes on NO2 concentrations, during the parts of the day when stability is unchanged, compared with the large NO2 level decrease ($\sim$ 15 $\mu$g m-3) at 4 pm for the simulations in which measured PBLH is used with and without the stability correction.

Page 17 Lines 648-654: "This is clearest outside the hours in which the stability correction has been applied, when large (∼300 $\mu$g m-3) measured and modelled mid-afternoon and nighttime PBLH discrepancies have negligible impact on simulated NO2 concentrations. The greater impact of PBL stability changes alone, however, is clearly evidenced by the ∼15 $\mu$g m-3 difference at 4 pm between simulations using measured PBLHs with and without the stability correction. This dominant influence of PBL stability is possibly related to the impact in the model configuration of near-surface traffic emissions and the exclusion of elevated point sources, with pollution dispersion from the latter more likely to be restricted by low PBLHs which would then further affect modelled NO2 levels."

Referee #2

In this paper, Biggart et al presented a street-level air quality simulation study for Beijing urban area using an urban air pollution dispersion and chemistry model ADMS-Urban. The predictions were evaluated against observations during winter campaign in 2016 in the same area. Several sensitivity tests were conducted to investigate possible reasons for the discrepancies between model and observations. Studies like this provide useful information on high-resolution air quality simulation in complex urban areas. The paper is generally well written. A few commons are provided below:

We thank the reviewer for their positive feedback.

1. The model domain for ADMS-Urban is 75 km x 90 km. Author mentioned that the model has street-level resolution, what is the exact resolution setting in the model in terms of metres? Does the resolution vary between different land uses, eg. Road and other areas? If they all use the same resolution as road, then it would require a very high computational demand.

We thank the reviewer for this comment and highlighting our need to precisely define 'street-level' resolution.

The reviewer is also correct to query whether the model resolution varies across the domain. To clarify, the pollutant concentration maps presented in Fig. 4 have been generated with both a regular grid of output points at ~150 m resolution as well as an array of receptor points added within and in the immediate vicinity of all individual road emission source segments. The additional receptor points increases the model resolution to < 10 m in regions containing dense distributions of explicit road sources, therefore enabling the sharp pollutant concentration variations adjacent to roads to be captured.

In Sections 3.2-3.7 modelled concentrations are compared with measurements recorded at monitoring network stations and the IAP field campaign site. For this work, as stated in Sect. 2.3 Line 308, modelled concentrations are output at exact locations by defining the coordinates of the measurement points in the model set-up. The computational demand of such simulations in which concentrations are output at a small number of locations is substantially lower than that required to produce the fully resolved maps in Fig. 4.

The manuscript has been altered to clarify the resolution of the concentration maps presented in Sect. 3.1. Page 9 Line 326-332 of the updated manuscript:

"In this study, the statistical evaluation of pollutant concentrations simulated at the exact coordinates of the measurement locations is complemented by street-scale resolution maps which more clearly illustrate the strong spatial heterogeneity of pollution levels across Beijing. Fully resolved PM2.5, NO2 and O3 concentration fields in central Beijing are simulated with a combination of regularly spaced receptor points at ~150 m and additional output points distributed within and in the immediate vicinity of all individual road emission source segments. The addition of emission source-oriented output points increases the model resolution to < 10 m across regions containing dense distributions of explicit road sources, therefore enabling the sharp pollutant concentration variations adjacent to roads to be captured."

2. Page 6 Line 225 "Measured concentrations at 12 of the 35 monitoring stations in Beijing were used to produce the background concentration." Is there a specific reason to exclude the other 23 stations?

This is an important point raised by the reviewer. We have used measurement data from the 12 stations in Beijing that are part of the national monitoring network run by the CNEMC. The remaining 23 stations making up the full network in Beijing were run by a different organisation and therefore may be subject to different data quality control procedures. Measurements from the 12 national stations we provided to all APHH-China participants. The manuscript has been updated to make this clearer, Page 6 Line 230-233 now reads:

"Measured concentrations at 12 national air quality monitoring stations, run by the China National Environmental Monitoring Center (CNEMC), the IAP field site and an additional site 60 km SE of Beijing, situated in the built-up Guangyang district of Langfang in Hebei province, are used to estimate this background concentration field."

3. Page 7 Line 245, agricultural emissions not included in the model simulation?

This is correct. The standard available (http://www.meicmodel.org/) MEIC v1.3 emissions include transportation, power, industry, residential and agricultural sectors. However, the 3 km MEIC emissions covering urban and suburban Beijing, provided by our APHH-China collaborators at Tsinghua University, exclude the agricultural sector. This is reasonable owing to the negligible amount of agriculture within our modelling domain. We have added this caveat to Page 7 Line 250 of the manuscript to clarify the omission of agricultural emissions:

"Note that the latter is not used in this study due to both the lack of farmland in urban Beijing and the negligible contributions to the pollutant species simulated in this study from agricultural emission sources (Qi et al. 2017)."

4. Page 7 Line 250, emissions for the industrial and residential sectors were distributed

based on Gross Domestic Product (GDP) and population density to grid-level resolution? What's the resolution for this? Is information of GDP and population density available at your grid level resolutions?

The industrial and residential source sectors (plus non-road transportation) in the MEIC emissions inventory, developed by Tsinghua University, were initially calculated at the provincial spatial scale using provincial level activity data (e.g energy consumption, fuel type, manufacturing technology, air pollution control devices) and emission factors. This information is largely unpublished but was collected by the MEIC development group from a range of databases (e.g Chinese Environmental Statistics, China's Pollution Source Census, China Energy Statistical Yearbook) (Qi et al. 2017). Emissions calculated for each province are then downscaled to county level and then to grid-scales of different resolutions using spatial proxies. County-level GDP, published in the China Statistical Yearbook by the National Bureau of Statistics (www.data.stats.gov.cn), and urban population are used for the industrial sector, with urban population and rural population used for the residential sector (Zheng et al. 2017). The population density data used for MEIC was generated by the Oak Ridge National Laboratory with the LandScan global population distribution model which generates global population distribution data at resolutions up to ~1 km (available for free download from www.landscan.ornl.gov). The MEIC emission inventory is produced by allocating emissions using the ~1 km resolution proxy data before aggregating to the coarser 3 km x 3 km resolution emissions used for this study (Qi et al. 2017).

The manuscript has been updated Page 7 Line 256-260:

"Industrial and residential sector emissions are calculated from provincial level activity data and emission factors (Zheng et al. 2017). Industrial emissions are then downscaled to the county level using GDP (National Bureau of Statistics, 2014), with both industry and residential emissions further distributed to grid level resolutions based on high resolution (~ 1 km) population density data (Oak Ridge National Laboratory, 2013) (Zheng et al. 2017)."

5. Table 3 provides the weighting factors for road emissions. Are the weighting factors the same for all pollutants that simulated in this study?

We thank the reviewer for this comment. Yes, the weighting factors provided in Table 3 are the same for all simulated pollutant species. In this study, the weighting factors for the allocation of gridded transportation emissions to individual road segments act as proxies for traffic volume (described on page 8 Line 290). For instance, motorways (weighting factor = 0.7) are expected to have substantially higher traffic volume than tertiary roads (weighting factor = 0.15). Further adjusting road emissions weighting factors for pollutant type would imply that the vehicle fleet composition on each road type is substantially different with different vehicle categories manufactured to varying emission standards resulting in some vehicles producing stronger or weaker emissions of particular pollutants than others. This may be an important consideration in Beijing due to, for instance, the strong temporal variation in location and volume of heavy duty diesel trucks (HDDTs). As discussed on Page 12 Line 452, HDDTs which often originate in provinces with less stringent emission standards flood into central Beijing at night following their daytime restrictions. Therefore, future work accounting for traffic volume, speed and composition variations on different road types would certainly be valuable but is outside the scope of this study.

We have added the following text to the updated manuscript. Page 8 Line 293: of the updated manuscript:

"Each road type weighting factor is applied equally to all pollutant species." Line 296:

"...This methodology is based on the assumption that traffic volume, speed and fleet composition are constant across all road type classes listed in Table 3."

Line 301:

"Additionally, Zhang et al. (2018) observed a greater proportion of vehicles with lower emission standards on roads outside the Fifth Ring Road."

6. Page 8 Line 290, the urban areas are more congested than suburban. This study used the same weightings factors for urban and suburban, which may underestimate the urban emissions. It is recommended to try different weighting factors for urban and suburban and test the impacts on simulation results.

We thank the reviewer for this suggestion. As we explain on Page 8 Line 297, substantial variations in traffic volume and speed with distance from Beijing's urban centre have previously been observed (Jing et al. 2016). However, this is largely accounted for by the underlying gridded transport sector emissions that are downscaled to grid level resolution using road network and vehicle kilometres travelled data (Zheng et al. 2014), producing a generally decreasing magnitude of transport emissions towards suburban Beijing. The application of different weighting factors on urban and suburban roads would imply that the greater congestion levels in urban areas results in, for example, a higher proportion of total traffic on primary versus tertiary roads compared to suburban areas. This seems like a reasonable hypothesis, however similarly to the previous suggestion to change weighting factors for different pollutants, reproducing the explicit traffic emissions inventory with varied weightings between urban and suburban areas and repeating simulations would require considerable extra computational effort than available. Adjusting weighting factors for pollutant type and urban/suburban areas would be interesting sensitivity studies for future work aiming to refine the production of explicit road emissions networks in urban areas where data-intensive bottom-up methodologies are not possible.

A sentence has been added to the manuscript (Sect. 4 Page 17 Line 676-679) to summarise the potential benefits of incorporating weighting factor variations:

"Future work could focus on refining the explicit road emissions network created here by testing the impact of adjusting weighting factors for different pollutants and across urban and suburban areas to better account for the impact of traffic congestion and vehicle type, such as HDDTs, on emissions along different road classifications."

7. Figure 16: suggest to change the color scheme for "Grids & Roads" and "Grid only". It is very hard to distinguish them on the map.

Thank you. The "Grids only" simulation results are now plotted in orange, with "Grids and Roads" in blue.

[revised manuscript text omitted]

Please also note the supplement to this comment:
https://www.atmos-chem-phys-discuss.net/acp-2019-783/acp-2019-783-AC1-supplement.pdf
* * *